# Retry Policy Gradients in Continuous Action Spaces

## Abstract

Retry-based objectives such as pass@K and max@K optimize the best return obtained from multiple sampled trajectories, and recent work has shown that they can promote exploration without explicit exploration bonuses. In discrete action spaces, ReMax was shown to do so by adapting to return uncertainty. In this work, we introduce pathwise derivative estimators for retry objectives and use them to extend ReMax to continuous action spaces. We study the resulting learning dynamics in a stylized fixed-state setting and show that, even with deterministic rewards, ReMax can promote higher policy stochasticity by reshaping the policy-gradient landscape. For a retry budget $M > 1$ and a sufficiently small policy scale, the ReMax gradient increases that scale whenever the policy mean is non-stationary. We also derive gradient bounds controlled by the distance of the best sampled action to the optimum: they vanish at the deterministic optimum, and the mean-gradient bound gives an upper envelope that decreases with the retry budget. We further show that Adam's adaptive normalization can mitigate the resulting gradient damping, depending on its numerical stabilization parameter. Empirically, we instantiate this objective as **ReMax Actor-Critic (ReMAC)**, an off-policy actor–critic algorithm that optimizes the ReMax objective using a pathwise derivative estimator. Our experiments show that ReMAC can promote higher policy entropy without entropy regularization and achieves performance comparable to SAC.

## 1 Introduction

Exploration remains a central challenge in reinforcement learning (RL) (Sutton & Barto, 2018). In continuous control, exploration is typically induced by stochastic policies combined with entropy regularization, as popularized by Soft Actor-Critic (SAC) (Haarnoja et al., 2018a; Christodoulou, 2019). Alternatively, noise injection—through either action-space or parameter-space perturbations—is widely used to encourage exploratory behavior (Fujimoto et al., 2018; Lillicrap et al., 2016; Plappert et al., 2018). Recent advances have further improved actor–critic methods through architectural refinements, such as the use of ensemble critics (Chen et al., 2021; Hiraoka et al., 2022).

Recently, retry-based objectives, which maximize the highest return achieved over multiple ($M \in \mathbb{N}$) sampled trajectories, have emerged as a compelling alternative formulation for RL in discrete action spaces (Koyamada et al., 2023; Nishimori et al., 2026; Tang et al., 2025; Walder & Karkhanis, 2025). In large language model (LLM) post-training, directly optimizing pass@K, which evaluates the success of the best response among $K$ generations, significantly improves output diversity (Tang et al., 2025; Chen et al., 2025; Walder & Karkhanis, 2025). Within episodic RL settings, the ReMax objective was introduced to explicitly account for the uncertainty of returns (Koyamada et al., 2023; Nishimori et al., 2026). Prior studies have revealed that ReMax naturally encourages exploration as an adaptive response to return uncertainty (Koyamada et al., 2023; Nishimori et al., 2026). Furthermore, Nishimori et al. (2026) demonstrated that ReMax can maintain high policy entropy by slowing down convergence, even in environments with deterministic rewards. However, these existing analyses are largely restricted to discrete action spaces, leaving the detailed theoretical properties of the ReMax objective incompletely understood.

In this study, we investigate the fundamental properties of the ReMax objective in continuous action spaces. Specifically, we identify intrinsic characteristics of the deterministic ReMax policy gradient, in both its *direction* and its *magnitude*, that can sustain policy stochasticity when the retry budget satisfies $M > 1$. In

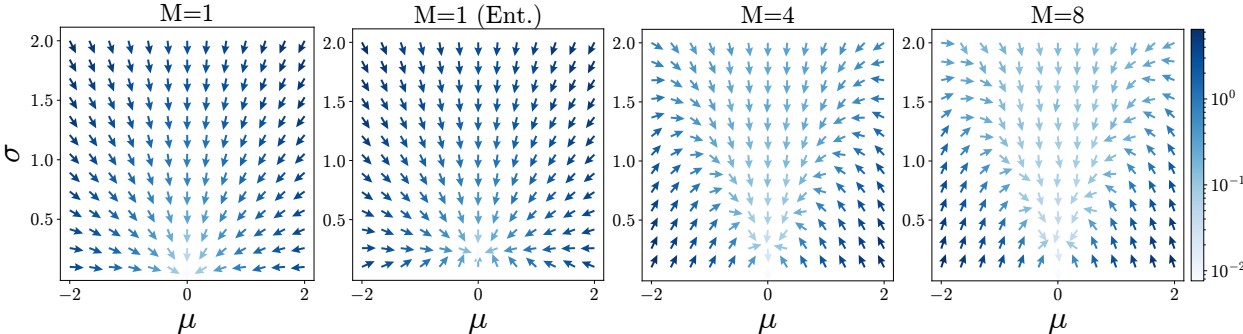

Figure 1: Vector field of normalized ReMax gradients for $M = 1, 4, 8$ over $\mu \in [-2, 2]$ and $\sigma \in [0.1, 2]$. We also plot the entropy-regularized standard RL ($M = 1$ (Ent.)). Arrow color indicates gradient norm (darker is larger). For larger $M$, the gradient increases $\sigma$ when $\mu$ is far from the optimum (bottom edge) and the gradient norm near the optimal mean (small $\mu$, nonzero $\sigma$) shrinks, slowing convergence. Unlike entropy regularization, ReMax always converges to the deterministic optimum $\mu = 0, \sigma = 0$.

a stylized fixed-state setting, we show that, for a sufficiently small policy scale, the ReMax gradient increases that scale whenever the policy mean is non-stationary. We also derive gradient bounds controlled by the distance of the best sampled action to the optimum: they vanish at the deterministic optimum, and the mean-gradient bound gives an upper envelope that decreases with $M$.

Fig. 1 illustrates this effect. It shows the vector field of ReMax gradients for a 1D Gaussian policy $a \sim \mathcal{N}(\mu, \sigma^2)$ with reward $r(a) = -a^2$. For larger $M$, when $\mu$ is far from the optimum and $\sigma$ is small (bottom edges), the gradient points toward increasing $\sigma$, whereas $M = 1$ (standard RL) always decreases $\sigma$. Conversely, when $\mu$ is near the optimal mean and $\sigma$ is large (top center), the gradient norm is attenuated (lighter color), slowing convergence. Unlike entropy regularization, ReMax does not change the optimal parameters $\mu = 0$ and $\sigma = 0$, whereas entropy regularization shifts the optimum to $\mu = 0$ and $\sigma > 0$. These properties yield an optimization trajectory in which policy entropy first increases and then gradually decreases, encouraging stochastic exploration before converging to the deterministic optimal policy. A detailed explanation of Fig. 1 is provided in Sec. 5.1, and a theoretical explanation for why these properties emerge is given in Sec. 5.2. Finally, the damping effect can be mitigated by Adam's adaptive gradient normalization (Kingma & Ba, 2015), the de facto standard optimizer in deep learning.

We then instantiate the ReMax objective in an off-policy actor–critic algorithm, which we call **ReM**ax **A**ctor-**C**ritic (**ReMAC**), with a pathwise derivative estimator. Empirically, we show that ReMAC achieves performance comparable to SAC while exhibiting increased entropy for larger $M$.

**Contributions.** Our main contribution is to identify intrinsic properties of the ReMax gradient, in both direction and magnitude, that can sustain policy stochasticity when $M > 1$. We illustrate this effect via a toy vector-field analysis and provide a theoretical explanation in the isotropic Gaussian policy setting. We then instantiate the ReMax objective in an off-policy actor–critic algorithm, ReMAC, which can be implemented with minimal modifications to SAC. Finally, we provide empirical evidence that ReMAC can increase policy entropy with $M > 1$ without entropy regularization and achieves performance comparable to SAC in continuous-control tasks.

## 2   Related Work

Extensive research has addressed the exploration-exploitation tradeoff in reinforcement learning (RL). Existing exploration strategies can be broadly categorized into two paradigms: *stochastic exploration* and *deep exploration*. Stochastic exploration maintains policy diversity to prevent premature convergence to suboptimal behaviors. In contrast, deep exploration seeks temporally coherent behaviors capable of uncovering informative, unvisited regions of the environment. A prevalent method for encouraging stochastic exploration is to introduce an entropy bonus into the policy optimization objective (Schulman et al., 2017; Haarnoja

et al., 2018a; Christodoulou, 2019; Lee et al., 2025; Nauman et al., 2024). This entropy regularization is a cornerstone of highly successful RL algorithms, such as Soft Actor-Critic (SAC), and remains a standard design choice in modern deep RL architectures (Nauman et al., 2024; Lee et al., 2025).

Conversely, deep exploration is typically driven by intrinsic rewards, uncertainty estimation, or structured noise injection. Standard intrinsic reward techniques include count-based exploration (Bellemare et al., 2016; Ostrovski et al., 2017; Lobel et al., 2023; Tang et al., 2017), prediction-error bonuses (Burda et al., 2019; Pathak et al., 2017), and information gain or curiosity signals (Houthooft et al., 2016; Sukhija et al., 2025; Schmidhuber, 2010). Alternatively, exploration can be promoted by modeling uncertainty in value estimates. For instance, ensemble-based methods approximate posterior sampling using multiple Q-functions (Osband et al., 2016; 2018; 2019; 2023; Zhang & Yao, 2019; Chen et al., 2021; Hiraoka et al., 2022), while related approaches leverage Langevin Monte Carlo updates (Ishfaq et al., 2024). Finally, structured noise injection, whether in the action space or parameter space, offers another mechanism for temporally extended exploration (Fujimoto et al., 2018; Lillicrap et al., 2016; Plappert et al., 2018; Fortunato et al., 2018).

Retry-based objectives offer a distinct paradigm for exploration. Rather than augmenting the reward function or injecting explicit noise into the policy, these objectives evaluate the maximum return achieved across multiple sampled trajectories. Prior work demonstrates that retry objectives successfully induce exploratory behavior in discrete action spaces (Koyamada et al., 2023; Nishimori et al., 2026; Walder & Karkhanis, 2025; Chen et al., 2025; Tang et al., 2025; Hamid et al., 2026). In the context of large language model (LLM) post-training, several recent studies optimize pass@K, or its generalization, max@K, to enhance the diversity of generated outputs (Tang et al., 2025; Chen et al., 2025; Walder & Karkhanis, 2025).

Within the retry-based framework, Koyamada et al. (2023) introduced a policy-gradient estimator for ReMax that relies on a resettable simulator. Later, Nishimori et al. (2026) developed a Q-function-based formulation enabling analytical gradient computation in discrete action spaces, demonstrating that ReMax can foster both stochastic and deep exploration. Specifically, deep exploration emerges when ReMax is paired with posterior sampling; notably, Tong et al. (2026) provided a finite-time regret analysis for this combination under a two-retry setting ($M = 2$). Conversely, stochastic exploration in ReMax has primarily been illustrated through a toy deterministic-reward bandit experiment, where it was observed that ReMax reshapes the gradient landscape and slows convergence, thereby encouraging exploration. However, this prior analysis remains restricted to discrete action spaces and lacks a rigorous, detailed examination of the gradient landscape.

In this paper, we investigate the emergence of *stochastic* exploration from ReMax in continuous action spaces with deterministic rewards, leaving the study of deep exploration for future work. Unlike the pass@K literature, which assumes explicit retries for a given prompt and directly observes outcomes, our setting deals with episodic returns where retries must be emulated via function approximation. Furthermore, while pass@K studies typically rely on sample-based REINFORCE-style estimators, we adopt a pathwise gradient estimator (Kingma & Welling, 2014; Parmas & Sugiyama, 2021; Jankowiak & Obermeyer, 2018; Rezende et al., 2014). This continuous-space pathwise estimator is not directly applicable to standard LLM post-training, which relies on non-differentiable, rule-based reward verifiers. Crucially, unlike traditional entropy regularization, ReMax alters the evaluation of rewards without introducing an auxiliary entropy bonus. Consequently, it naturally avoids bonus-induced bias at the target optimum and eliminates the need to decay bonus coefficients.

## 3 Preliminaries

**Reinforcement learning (RL).** We consider the reinforcement learning (RL) problem (Sutton & Barto, 2018), formulated as a Markov decision process (MDP), $\mathcal{M} = (\mathcal{S}, \mathcal{A}, \mathcal{P}, p_0, \mathcal{R}, \gamma)$, where $\mathcal{S}$ is the state space, $\mathcal{A} \subseteq \mathbb{R}^d$ is the continuous action space with dimension $d$, $\mathcal{P}$ is the transition dynamics, $p_0$ is the initial state distribution, $\mathcal{R} : \mathcal{S} \times \mathcal{A} \to \mathbb{R}$ is the reward function, and $\gamma \in [0, 1)$ is the discount factor. The goal is to train a policy $\pi(\cdot \mid s)$ that maximizes the expected discounted reward $\mathbb{E}_\pi [R_0]$, where $R_0 = \sum_{h=0}^{\infty} \gamma^h r(s_h, a_h)$ is the discounted return and $\mathbb{E}_\pi [\cdot]$ denotes the expectation over the trajectory $(s_0, a_0, r_0, s_1, a_1, r_1, \ldots)$ under policy $\pi$. We define the value function of the policy $\pi$ as $V^\pi(s) = \mathbb{E}_\pi [R_0 \mid s_0 = s]$ and the Q-function as $Q^\pi(s, a) = \mathbb{E}_\pi [R_0 \mid s_0 = s, a_0 = a]$.

# 4    ReMax Objective and its Gradient Estimation

This section introduces the ReMax objective and describes how we estimate its policy gradient.

## 4.1    ReMax Objective

Optimizing returns over multiple trajectories can promote exploration without explicit exploration bonuses (Walder & Karkhanis, 2025; Nishimori et al., 2026; Koyamada et al., 2023). In episodic RL, directly optimizing such retry-based trajectory objectives often assumes the ability to reset the environment to arbitrary states (Koyamada et al., 2023), which is typically infeasible in continuous-control tasks. We therefore adopt the Q-function-based deterministic ReMax formulation (Nishimori et al., 2026). Given a Q-function $Q : \mathcal{S} \times \mathcal{A} \to \mathbb{R}$, the (deterministic) ReMax objective is defined as

$$\mathcal{J}^M(\pi, Q) := \mathbb{E}_{s \sim p_0}\left[\mathcal{J}^M(\pi, Q, s)\right], \quad \text{where} \quad \mathcal{J}^M(\pi, Q, s) := \mathbb{E}_{a_{1:M} \overset{\text{i.i.d.}}{\sim} \pi(\cdot|s)}\left[\max_{m=1:M} Q(s, a_m)\right] \tag{1}$$

where $M \in \mathbb{N}$ is the retry budget and $\pi$ is the policy. When $M = 1$ and $Q = Q^\pi$, this objective $\mathcal{J}^1(\pi, Q^\pi)$ reduces to the standard RL objective.

Importantly, the ReMax objective preserves the deterministic optimal policy. Since an MDP admits an optimal deterministic policy under standard conditions (Sutton & Barto, 2018), assume that there exists a deterministic optimal policy $\pi^*$ satisfying $Q^{\pi^*} = Q^*$ and $\pi^*(s) \in \arg\max_{a \in \mathcal{A}} Q^*(s, a)$ for every state $s \in \mathcal{S}$. Then $\pi^* \in \arg\max_\pi \mathcal{J}^M(\pi, Q^\pi, s)$ for any finite $M \geq 1$. Indeed, for any state $s$ and policy $\pi$,

$$\mathbb{E}_{a_{1:M} \overset{\text{i.i.d.}}{\sim} \pi(\cdot|s)}\left[\max_{m=1:M} Q^\pi(s, a_m)\right] \leq \sup_a Q^*(s, a) = \mathbb{E}_{a_{1:M} \overset{\text{i.i.d.}}{\sim} \pi^*(\cdot|s)}\left[\max_{m=1:M} Q^*(s, a_m)\right],$$

where $Q^*(s, a) = \sup_\pi Q^\pi(s, a)$. Therefore, we have $\mathcal{J}^M(\pi, Q^\pi) \leq \mathcal{J}^M(\pi^*, Q^{\pi^*})$. In contrast, entropy-regularized objectives shift the optimum toward higher-entropy policies and hence bias the optimum.

## 4.2    Gradient Estimation of the ReMax Objective

We consider optimizing Eq. (1) using policy gradients. Nishimori et al. (2026) showed that in discrete action spaces, both the objective and its gradient can be computed exactly. This approach does not extend to continuous action spaces because it would require sorting Q-values over an uncountable action set. We therefore use sample-based estimators (Parmas et al., 2018; Parmas, 2018). Let $\pi_\theta$ be a policy parameterized by $\theta \in \mathbb{R}^p$. For a state $s$, we sample $a_{1:B} \overset{\text{i.i.d.}}{\sim} \pi_\theta(\cdot \mid s)$ with $B \geq M$ and compute $q_i = Q(s, a_i)$. In practice, $Q$ is approximated by a neural network $Q_\phi$ with parameters $\phi$. We consider two gradient estimators.

**Likelihood ratio (LR) gradient estimator.** The likelihood-ratio estimator approximates the gradient of $\mathbb{E}_{a \sim \pi_\theta}[f(a)]$ $(f : \mathcal{A} \to \mathbb{R})$ as $\mathbb{E}_{a \sim \pi_\theta}[\nabla_\theta \log \pi_\theta(a \mid s)f(a)]$, also known as the REINFORCE estimator (Williams, 1992). Because Eq. (1) involves a maximum over $M$ coupled action samples, the single-sample LR form above is not directly applicable as written. Nevertheless, Walder & Karkhanis (2025) derived an LR-style estimator for deterministic ReMax: $\hat{g} = \frac{1}{B}\sum_{i=1}^{B} \nabla_\theta \log \pi_\theta(a_i \mid s)\widehat{\text{Adv}}_i$, where $\widehat{\text{Adv}}_i$ is an advantage estimate. We refer the reader to Walder & Karkhanis (2025) for details.

**Reparameterization (RP) gradient estimator.** Alternatively, we can estimate the gradient via the reparameterization trick (Kingma & Welling, 2014). Assume $\pi_\theta$ is Gaussian with mean $\mu$ and standard deviation $\sigma$ $(\theta = (\mu, \sigma))$. Actions are sampled as $a_i(\mu, \sigma) = \mu + \sigma\xi_i$, where $\xi_i \sim \mathcal{N}(0, 1)$. Thus $\frac{\partial a_i}{\partial \mu} = 1$ and $\frac{\partial a_i}{\partial \sigma} = \xi_i$. For higher-dimensional actions, see Rezende et al. (2014). For a $d$-dimensional action, $\mu, \sigma \in \mathbb{R}^d$ and $a_i = \mu + \sigma \odot \xi_i$ with $\xi_i \sim \mathcal{N}(0, I_d)$; the $\xi_{1:B}$ of Eq. (4) are then $B$ noise vectors in $\mathbb{R}^d$. When $Q$ is differentiable, we can compute $\nabla_{a_i} Q(s, a_i)$ and backpropagate through $q_i$. Walder & Karkhanis (2025) proposed an unbiased estimator of Eq. (1) (for the given $Q$) from $q_{1:B}$ (sorted in ascending order with ties broken randomly):

$$\rho^M(q_{1:B}) := \frac{1}{\binom{B}{M}}\sum_{i=1}^{B} w_i q_i, \tag{2}$$

where $w_i = \binom{i-1}{M-1}$ counts the number of size-$M$ subsets for which the $i$-th sorted action is the best; the factor $\binom{B}{M}^{-1}$ converts this count into a probability. Because $\rho^M$ is linear in $q_{1:B}$, $\nabla_\theta \rho^M(q_{1:B})$ can be obtained by automatic differentiation. Walder & Karkhanis (2025) did not explore this RP estimator because their focus was LLM post-training, where rewards are typically non-differentiable.

**Our choice.** The two estimators have different trade-offs. LR gradients can remain unbiased with raw returns, which suits on-policy learning, but they often suffer from high variance (Greensmith et al., 2004). RP gradients require a differentiable Q-function but typically (but not always) yield lower-variance estimates (Parmas et al., 2018). In continuous control, off-policy actor–critic methods (Degris et al., 2012; Haarnoja et al., 2018a; Fujimoto et al., 2018) are widely used for their sample efficiency. These methods rely on replay buffers that store transitions from past behavior policies and update both the Q-function and policy off-policy. As a result, obtaining raw-return targets for arbitrary state–action pairs is difficult, which weakens the main advantage of LR estimators. Thus, in this setting, LR offers limited benefits while retaining high variance, whereas RP estimators are standard and effective. We therefore adopt the RP estimator.

# 5 Gradient Analysis

We analyze how the ReMax objective shapes policy gradients. Even with deterministic rewards, ReMax can promote higher policy stochasticity by reshaping the gradient landscape. In a stylized fixed-state setting, we show that, for $M > 1$ and a sufficiently small policy scale (the standard deviation $\sigma$ of the Gaussian policy), the ReMax gradient increases that scale whenever the policy mean is non-stationary. We also derive gradient bounds controlled by the distance of the best sampled action to the optimum: they vanish at the deterministic optimum, and the mean-gradient bound gives an upper envelope that decreases with $M$. We first illustrate both effects in a toy vector field (Sec. 5.1), then establish them (Sec. 5.2), and finally show that Adam's normalization can mitigate the damping effect (Sec. 5.3), offering practical guidance for continuous control.

**Notation.** The action space is $\mathcal{A} \subseteq \mathbb{R}^d$, and $d$ denotes the action dimension. The policy is an isotropic Gaussian $\mathcal{N}(\mu, \sigma^2 I_d)$ with mean $\mu \in \mathbb{R}^d$ and policy scale $\sigma \in \mathbb{R}_{>0}$, a single scalar shared across action dimensions. Accordingly, $\nabla_\mu$ is a gradient with respect to a vector, measured in the Euclidean norm $\|\cdot\|$, and $\partial_\sigma$ is a derivative with respect to a scalar, so $\partial_\sigma J^M$ is a real number. Action samples are written $A_m$ and reparameterization noise $\xi_m \sim \mathcal{N}(0, I_d)$. We also write $Z_1, \ldots, Z_M \overset{\text{i.i.d.}}{\sim} \mathcal{N}(0, 1)$ and $\beta_M := \mathbb{E}[\max_{m \leq M} Z_m]$, so that $\beta_1 = 0$ and $\beta_M > 0$ for $M \geq 2$.

The analyzed objective is the fixed-state, fixed-critic analogue of the actor objective that ReMAC differentiates. For a state $s$ and a fixed critic $Q$, write $c(a) := -Q(s, a)$; the object we study is then the minimization of a deterministic cost $c : \mathbb{R}^d \to \mathbb{R}$ of the action alone, so the state dependence of $Q(s, a)$ is absorbed into the choice of $c$. ReMAC instead differentiates an expectation over states drawn from the replay buffer, with a critic that is simultaneously being fit and a state-dependent diagonal scale. The gradient with respect to the actual network parameters $\theta$ follows by the chain rule through the Jacobian of $(\mu_\theta(s), \sigma_\theta(s))$, so its norm is bounded by the operator norm of that Jacobian times the norms of the $\mu$- and $\sigma$-gradients; the mechanisms studied here therefore transfer to $\theta$ as long as this Jacobian stays bounded, and for multiple states the objective averages the fixed-state landscape over the state distribution.

To illustrate this effect, we consider a one-dimensional ($d = 1$) Gaussian policy with mean $\mu \in \mathbb{R}$ and standard deviation $\sigma > 0$, i.e., $a \sim \mathcal{N}(\mu, \sigma^2)$. We use the quadratic reward $r(a) = -a^2$.

## 5.1 Vector field analysis

In Fig. 1, we visualize the vector field of the RP gradient estimator (Sec. 4.2) with batch size $B = 16$ over $\mu \in [-2, 2]$ and $\sigma \in [0.1, 2]$, for $M \in \{1, 4, 8\}$. For reference, we also plot the entropy-regularized objective for $M = 1$, $J_{\text{ent}}(\mu, \sigma) = \mathbb{E}_{a \sim \mathcal{N}(\mu, \sigma^2)}[r(a)] + \alpha \mathcal{H}(\mathcal{N}(\mu, \sigma^2))$ with $\alpha = 0.5$, where $\mathcal{H}(\mathcal{N}(\mu, \sigma^2)) = \frac{1}{2} \log(2\pi e \sigma^2)$. For $M = 1$, the gradient with respect to $\sigma$ is always negative. For $M > 1$, ReMax exhibits two exploration-promoting properties:

- **Entropy Increase (direction):** When the mean is far from the optimum and $\sigma$ is small, the gradient increases $\sigma$, inducing stochastic exploration.

- **Gradient Damping (magnitude):** As $M$ increases, the gradient norm near the optimal mean (small $\mu$ and large $\sigma$) decreases, slowing convergence and helping sustain larger entropy.

More details are provided in App. B.1. ReMax converges to the deterministic optimum $(\mu, \sigma) = (0, 0)$, whereas entropy regularization converges to $(0, 0.5)$. Together, these effects keep entropy high during optimization: it first increases due to the directional effect and then decreases gradually due to damping.

## 5.2 Theoretical Explanation

We now provide intuition for how ReMax with $M > 1$ shapes exploration dynamics in high-dimensional action spaces. The vector-field analysis in Sec. 5.1 is one-dimensional, but the underlying mechanism is not specific to one dimension. When the policy scale is small, the selected action is determined by the projection of Gaussian perturbations onto the local improvement direction. Thus, ReMax favors samples that move the action toward lower cost, yielding a positive scale gradient when the current policy mean is not stationary. When the policy mean is near the optimum but the policy scale remains nonzero, selecting the best action among $M$ samples increases the chance that the selected action lies close to the optimum, which can attenuate the gradient magnitude. These two effects correspond to the entropy-increase and gradient-damping behaviors observed in the vector field.

**Setup and assumptions.** Let $\mu \in \mathbb{R}^d$ and $\sigma > 0$ denote the mean and scalar scale of an isotropic Gaussian policy. For $m = 1, \ldots, M$, let

$$A_m = \mu + \sigma \xi_m, \qquad \xi_m \overset{\text{i.i.d.}}{\sim} \mathcal{N}(0, I_d).$$

We consider deterministic rewards of the form $r(a) = -c(a)$ with $c : \mathbb{R}^d \to \mathbb{R}$. For $M \geq 1$, define $J^M(\mu, \sigma) = \mathbb{E}\left[\max_{m=1,\ldots,M} -c(A_m)\right]$. Equivalently, $J^M(\mu, \sigma) = -\mathbb{E}\left[\min_{m=1,\ldots,M} c(A_m)\right]$. Let $m^\star \in \arg\min_{m=1,\ldots,M} c(A_m)$ denote the selected action. These assumptions are introduced for analytical tractability rather than to model deep RL environments globally, allowing us to study how ReMax shapes the local gradient landscape.

**Assumption 1** (Smoothness). $c$ is differentiable and $L$-smooth, i.e., $\|\nabla c(a) - \nabla c(b)\| \leq L\|a - b\|$ for all $a, b \in \mathbb{R}^d$.

**Assumption 2** (Non-degenerate selection). There exists $\bar{\sigma} > 0$ such that, for every $\sigma \in (0, \bar{\sigma})$, the random variable $c(\mu + \sigma \xi)$ has no atoms for $\xi \sim \mathcal{N}(0, I_d)$.

This assumption ensures that the selected sample is almost surely unique. It holds, for example, when every level set $\{a : c(a) = t\}$ has Lebesgue measure zero. As shown in App. A, this condition is automatically satisfied under Assumptions 1, 3, and 4.

**Assumption 3** (Centered optimum). The cost is normalized so that $c(0) = 0$ and $\nabla c(0) = 0$.

**Assumption 4** (Strong convexity). $c$ is $\lambda$-strongly convex, i.e.,

$$c(b) \geq c(a) + \nabla c(a)^\top (b - a) + \frac{\lambda}{2}\|b - a\|^2$$

for all $a, b \in \mathbb{R}^d$.

**Assumption 5** (Second-order regularity). $c$ is twice continuously differentiable.

These assumptions are separated to make clear which part of the analysis relies on which regularity condition. The entropy-increase result does not require convexity: it relies on global $L$-smoothness and non-degenerate sample selection, and its small-scale limit is determined by the local gradient $\nabla c(\mu)$. The entropy-decrease result for $M = 1$ additionally uses second-order regularity and strong convexity. The gradient-damping result uses smoothness, strong convexity, and the normalization that the optimum is centered at the origin. These assumptions are not intended to be the weakest possible conditions. Rather, they provide a clean first setting

in which the two mechanisms observed in the vector field can be isolated and proved rigorously. In particular, smoothness and strong convexity are standard assumptions in first-order optimization theory (Nesterov, 2013; Bubeck, 2015), while the non-degenerate selection condition is a technical condition used to handle the best-of-$M$ operator. Thus, although these assumptions do not fully reflect practical deep-RL settings with neural networks—in particular, we analyze the actor's objective with $Q$ held fixed, rather than the coupled dynamics in which the critic is simultaneously re-fit—they are appropriate for a first theoretical analysis of the local mechanisms induced by ReMax.

**Entropy increase effect for $M > 1$.** We first show that ReMax can increase the policy scale when the current mean is not stationary and the scale is small. This result does not require global convexity. It only uses the local first-order geometry of the cost around the current policy mean.

**Proposition 1** (Entropy Increase Effect). Assume Assumptions 1 and 2. Let $M \geq 2$ and suppose that $\nabla c(\mu) \neq 0$. Then, there exists $\sigma_0 > 0$ such that

$$\partial_\sigma J^M(\mu, \sigma) > 0 \qquad \text{for all } \sigma \in (0, \sigma_0).$$

The proof is given in App. A.1. The intuition is that ReMax can exploit upward fluctuations in reward across multiple samples. Since $r(a) = -c(a)$, this is equivalent to selecting a sample with a downward fluctuation in cost. Let $g = \nabla c(\mu)$. For small $\sigma$, the cost of each sampled action is locally approximated as $c(\mu + \sigma \xi_m) = c(\mu) + \sigma g^\top \xi_m + O(\sigma^2 \|\xi_m\|^2)$. Thus, to first order, the selected action is the one with the smallest projection $g^\top \xi_m$, or equivalently the largest reward fluctuation $-g^\top \xi_m$. For a single sample, this fluctuation has zero mean. With $M \geq 2$, however, taking the maximum over samples creates a positive expected upward fluctuation, since $\mathbb{E}[\max_m (-g^\top \xi_m)] = -\mathbb{E}[\min_m g^\top \xi_m] = \|\nabla c(\mu)\| \beta_M > 0$, where $\beta_M = \mathbb{E}[\max_{m \leq M} Z_m]$ and $Z_1, \ldots, Z_M \overset{\text{i.i.d.}}{\sim} \mathcal{N}(0, 1)$. Therefore, when $\sigma$ is small, increasing the policy scale makes these favorable fluctuations more available, yielding a positive scale gradient. This is the high-dimensional analogue of the entropy-increase direction observed in Sec. 5.1.

The result is local in the policy scale. It guarantees a positive scale derivative only for sufficiently small $\sigma$, and the proof gives the limiting value $\lim_{\sigma \downarrow 0} \partial_\sigma J^M(\mu, \sigma) = \beta_M \|\nabla c(\mu)\|$ (App. A.1). It does not determine the sign of $\partial_\sigma J^M$ at an arbitrary $\sigma$, nor does it relate the sign to stationarity of the mean in both directions.

**Entropy decrease for $M = 1$.** The previous effect contrasts with the standard single-sample objective. When $M = 1$, increasing the policy scale only injects noise into the cost, and the scale gradient is always negative under strong convexity.

**Proposition 2** (Entropy Decrease for $M = 1$). Assume Assumptions 1, 4, and 5. For $M = 1$, $\mu \in \mathbb{R}^d$, and $\sigma > 0$,

$$\partial_\sigma J^1(\mu, \sigma) \leq -(\lambda d)\, \sigma < 0.$$

The proof is given in App. A.2. This result explains why the $M = 1$ vector field consistently shrinks the policy scale. Without best-of-$M$ selection, the objective directly penalizes variance around the current mean.

**Gradient damping effect for $M > 1$.** We next bound the gradient magnitudes by the expected distance of the best sampled action to the optimum.

**Proposition 3** (Gradient Damping). Assume Assumptions 1, 3, and 4. Let $M \geq 1$ and $\sigma > 0$. Then, the gradients exist and satisfy

$$\|\nabla_\mu J^M(\mu, \sigma)\| \leq L \sqrt{\frac{L}{\lambda}} \, \mathbb{E}\left[\min_{m=1,\ldots,M} \|A_m\|\right],$$

and

$$|\partial_\sigma J^M(\mu, \sigma)| \leq L \sqrt{\frac{L}{\lambda}} \, \mathbb{E}\left[\min_{m=1,\ldots,M} \|A_m\| \|\xi_{m^\star}\|\right].$$

The proof is given in App. A.3.

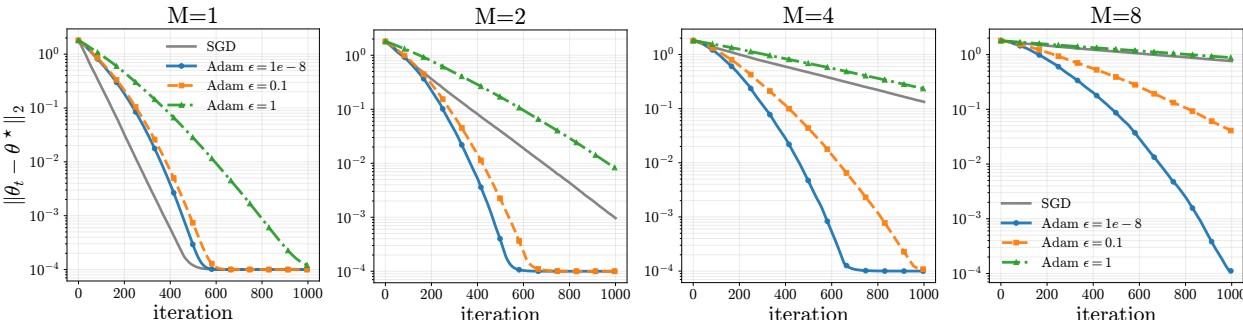

Figure 2: The distance to the optimal parameter $(\mu, \sigma) = (0, 0)$ for Adam with different $\varepsilon$ values and for SGD. As the vector field indicates, the convergence speed slows as $M$ increases in every case. However, with smaller $\varepsilon$ the convergence near the optimum remains fast, whereas larger $\varepsilon$ aligns the trajectory with the SGD path that follows the vector field.

The two expectations can be bounded in terms of $\mu$, $\sigma$, and $M$, which makes the dependence on the retry budget explicit. Define the *retry constants* $\kappa_{d,M} := \mathbb{E}\left[\min_{m \leq M} \|\xi_m\|\right]$, $\tau_{d,M} := \mathbb{E}\left[\min_{m \leq M} \|\xi_m\|^2\right]$, and $\nu_{d,M} := (\mathbb{E}\left[\max_{m \leq M} \|\xi_m\|^2\right])^{1/2}$, which depend only on $d$ and $M$.

**Corollary 1** (Explicit form of the damping). Under the assumptions of Proposition 3,

$$\|\nabla_\mu J^M(\mu, \sigma)\| \leq L\sqrt{\frac{L}{\lambda}}\left(\|\mu\| + \sigma\,\kappa_{d,M}\right), \qquad |\partial_\sigma J^M(\mu, \sigma)| \leq L\sqrt{\frac{L}{\lambda}}\left(\|\mu\| + \sigma\sqrt{\tau_{d,M}}\right)\nu_{d,M}.$$

Moreover $\kappa_{d,M}$ and $\tau_{d,M}$ are strictly decreasing in $M$, with $\kappa_{d,M} \leq \Gamma(1 + \frac{1}{d})(c_d M)^{-1/d} + \sqrt{d}\,(1 - c_d)^{M-1}$ for $c_d = e^{-1/2}/(2^{d/2}\Gamma(\frac{d}{2} + 1))$, while $\nu_{d,M} \leq (1 + (\sqrt{d} + \sqrt{2\log M})^2)^{1/2}$ grows at most on the order of $\sqrt{\log M}$.

The proof is given in App. A.4. Both bounds increase in $\|\mu\|$ and $\sigma$, so they guarantee that the gradient norm converges to zero as $(\mu, \sigma)$ approaches the deterministic optimum $(0, 0)$. This is a limiting statement, and it does not imply that the gradient norm decreases monotonically along an optimization trajectory. At any fixed $(\mu, \sigma)$, the $\mu$-bound is decreasing in $M$ through $\kappa_{d,M}$: the corollary provides an $M$-decreasing upper envelope for the mean-gradient norm. It does not establish that the gradient norm itself decreases with $M$. The $\sigma$-bound involves both $\sqrt{\tau_{d,M}}$, which decreases with $M$, and $\nu_{d,M}$, which increases with $M$. For the scale gradient, the nearest-sample term decreases with $M$, but the additional factor $\nu_{d,M}$ prevents a general monotonicity conclusion for the full bound.

These bounds capture the "forgiveness" of the minimum operator. In standard RL ($M = 1$), a large policy scale near the optimal mean produces poor actions and strong negative scale gradients that rapidly shrink $\sigma$. In contrast, ReMax evaluates only the best action. When $\mu$ is near 0, increasing $M$ raises the probability that at least one action lies close to the optimum, reducing the selected deviation and damping gradients. When the policy mean is far from stationary and $\sigma$ is small, the projection effect in Proposition 1 dominates and increases the policy scale. Together, these effects can keep entropy high during optimization: it first increases due to the directional effect and then decreases gradually due to damping.

### 5.3 Effect of Adam's normalization

Although the vector-field analysis suggests that ReMax with $M > 1$ promotes stochastic exploration, the actual optimization trajectory need not follow the raw vector field. With Adam (Kingma & Ba, 2015), adaptive normalization can counteract gradient damping, leading to faster convergence and faster entropy collapse. Adam updates parameters as $\theta_{t+1} = \theta_t - \alpha\frac{\hat{m}_t}{\sqrt{\hat{v}_t} + \varepsilon}$, where $\hat{m}_t$ and $\hat{v}_t$ are the bias-corrected first and second moments. If $\varepsilon$ is small, $\frac{\hat{m}_t}{\sqrt{\hat{v}_t} + \varepsilon} \approx \frac{\hat{m}_t}{\sqrt{\hat{v}_t}}$, effectively normalizing gradients and reducing damping. If instead $\sqrt{\hat{v}_t} \ll \varepsilon$, the update is approximately $\hat{m}_t/\varepsilon$, so the large-$\varepsilon$ regime approaches *momentum* SGD with effective learning rate $\alpha/\varepsilon$—not plain SGD—and the gradient magnitude is transmitted to the parameters. In the toy experiment below, $\varepsilon = 1$ empirically tracks the plain-SGD baseline, but the value 1 has no special

---

**Algorithm 1** ReMAC

---

**Require:** Retry parameter $M$, action batch size $B$, discount $\gamma$, learning rates $\alpha, \beta$, target critic update rate $\tau$.

1: Initialize policy $\pi_\theta$, critics $Q_{\phi_1}, Q_{\phi_2}$, target critics $Q_{\bar{\phi}_1}, Q_{\bar{\phi}_2}$, and replay buffer $\mathcal{D}$
2: **for** each iteration $t$ **do**
3:     **for** each environment step **do**
4:         $a \sim \pi_\theta(\cdot|s),\ (r,s') \sim \mathcal{P}(s,a)$                             ▷ interact with environment
5:         $\mathcal{D} \leftarrow \mathcal{D} \cup \{(s,a,r,s')\}$                                   ▷ store transition
6:     **end for**
7:     **for** each gradient step **do**
8:         $(s,a,r,s') \sim \mathcal{D},\ a' \sim \pi_\theta(\cdot|s')$                ▷ sample minibatch and next action
9:         $y \leftarrow r + \gamma \min(Q_{\bar{\phi}_1}(s',a'), Q_{\bar{\phi}_2}(s',a'))$                   ▷ target value
10:        $\phi_j \leftarrow \phi_j - \alpha \nabla_{\phi_j}(Q_{\phi_j}(s,a) - y)^2,\ j \in \{1,2\}$          ▷ update critics
11:        $a_{1:B} \sim \pi_\theta(\cdot|s),\ q_i \leftarrow \min_{j=1,2} Q_{\phi_j}(s,a_i) \forall i \in \{1,2,\ldots,B\}$   ▷ sample actions and evaluate Q
12:        $\theta \leftarrow \theta + \beta \nabla_\theta \rho^M(q_{1:B})$                               ▷ update policy
13:        $\bar{\phi}_j \leftarrow \tau\phi_j + (1-\tau)\bar{\phi}_j,\ j \in \{1,2\}$                  ▷ update target critics
14:     **end for**
15: **end for**

---

theoretical status. Varying $\varepsilon$ rescales the denominator but never removes it; an ablation that deletes it entirely is reported in App. B.2.

To illustrate this effect, we optimize $a \sim \mathcal{N}(\mu, \sigma^2)$ with $r(a) = -a^2$ using Adam with $\varepsilon \in \{10^{-8}, 10^{-1}, 1.0\}$. We set $\alpha = 0.01$, $B = 64$, $T = 1000$, and $M \in \{1,2,4,8\}$, starting from $(\mu, \sigma) = (-1.5, 1.0)$. Fig. 2 shows the distance to $(0,0)$ for Adam and SGD. As expected, convergence slows as $M$ increases. However, smaller $\varepsilon$ mitigates damping: with $M = 4, 8$, $\varepsilon = 10^{-8}$ converges much faster than SGD, whereas $\varepsilon = 1.0$ aligns with SGD. These results suggest a practical guideline. Adam benefits from the directional entropy-increase effect, but with the default $\varepsilon = 10^{-8}$, damping is weakened. Increasing $\varepsilon$ restores damping but reduces the effective step size. Jointly tuning $\varepsilon$ and the learning rate may therefore better control optimization dynamics during training. More details are provided in App. B.2.

# 6   ReMax Actor-Critic Algorithm

Having identified the policy-stochasticity effects of ReMax, we instantiate it in a simple off-policy actor–critic algorithm with a critic $Q_\phi$ and a stochastic policy $\pi_\theta$. The critic is trained by temporal-difference regression on transitions sampled from a replay buffer $\mathcal{D}$, while the actor is optimized using ReMax computed from batches of action samples.

**Critic update.** Given a transition $(s,a,r,s') \sim \mathcal{D}$, we define the target as $y = r + \gamma Q_{\bar{\phi}}(s',a')$, where $a' \sim \pi_\theta(\cdot \mid s')$ and $Q_{\bar{\phi}}$ is a target critic. The critic is updated by minimizing the squared Bellman error:

$$L_Q(\phi) := \mathbb{E}_{(s,a,r,s') \sim \mathcal{D}}\big[(Q_\phi(s,a) - y)^2\big]. \tag{3}$$

**Actor update.** For policy optimization, we use a tanh-squashed Gaussian policy $\pi_\theta(\cdot \mid s)$, as in SAC, whose mean $\mu_\theta(s) \in \mathbb{R}^d$ and diagonal scale $\sigma_\theta(s) \in \mathbb{R}^d_{>0}$ are neural-network outputs; unlike the scalar $\sigma$ of Sec. 5, the scale is a state-dependent vector. We sample $B$ independent noise vectors $\xi_i \overset{\text{i.i.d.}}{\sim} \mathcal{N}(0, I_d)$ and construct the pre-squash actions as

$$\tilde{a}_i = \mu_\theta(s) + \sigma_\theta(s) \odot \xi_i, \qquad a_\theta(\xi_i) = \tanh(\tilde{a}_i), \qquad i = 1, \ldots, B,$$

where $\odot$ is the elementwise product. We then evaluate the Q-values as $Q_\phi(s, a_\theta(\xi_{1:B})) := (Q_\phi(s, a_\theta(\xi_1)), \ldots, Q_\phi(s, a_\theta(\xi_B)))$. The actor loss is

$$L_{\text{actor}}^M(\theta) := -\mathbb{E}_{s \sim \mathcal{D},\ \xi_{1:B} \overset{\text{i.i.d.}}{\sim} \mathcal{N}(0, I_d)}\big[\rho^M(Q_\phi(s, a_\theta(\xi_{1:B})))\big]. \tag{4}$$

We optimize the policy parameters by automatic differentiation of $L_{\text{actor}}^M(\theta)$. We call this algorithm **ReMax Actor-Critic** (**ReMAC**); the full procedure is summarized in Alg. 1.

**Remarks on the implementation.** To reduce overestimation, we use clipped double Q-learning (Haarnoja et al., 2018a; Fujimoto et al., 2018) with two critics and take the minimum in both the target and actor loss. In practice, we obtain ReMAC by removing the "soft" components of Soft Actor-Critic (SAC) (Haarnoja et al., 2018a)—namely, entropy bonuses in the critic targets and temperature tuning—and replacing the actor loss with our ReMax loss.

This is a minimal off-policy actor–critic instantiation of ReMax, and many extensions are possible. One promising direction is to maintain a posterior over the Q-function (Osband et al., 2016; 2018; 2019), making return uncertainty explicit as in ReMax (Koyamada et al., 2023; Nishimori et al., 2026). This may enable *deeper*, more structured exploration, which is crucial in sparse-reward settings.

## 7 Experiments

The primary goal of our empirical evaluation is *not* to establish ReMAC as a new state-of-the-art exploration method for continuous control. Instead, we treat this section as a proof-of-concept evaluation and ask the following two questions.

1. Does ReMAC achieve performance comparable to SAC in continuous-control tasks?

2. Does ReMAC exhibit higher policy entropy for $M > 1$ in practical deep-RL training?

**Setup.** We considered six continuous-control tasks from Brax (Freeman et al., 2021): Ant, HalfCheetah, Hopper, Reacher, Swimmer, and Walker2d, together with HumanoidStandup ($d = 17$ action dimensions), the highest-action-dimensional task for which `rejax` provides a tuned configuration. Agents were trained for 3 million environment steps. We report the average return over 128 evaluation episodes, together with the standard error across 10 random seeds.

**Baselines.** We used SAC (Haarnoja et al., 2018a) and PPO (Schulman et al., 2017) as baselines to verify that ReMAC maintains competitive performance. Our implementation built on `rejax` (Liesen et al., 2024), which provides vectorized training pipelines for SAC and PPO with tuned configurations for each Brax task; we adopted these configurations for the baselines. SAC uses automatic temperature tuning (Haarnoja et al., 2018b): the temperature is a learned parameter, initialized at $\alpha = 1$ and adjusted to hold the policy entropy at the target $-\dim(\mathcal{A})$. We evaluate ReMAC with retry budgets $M \in \{1, 2, 4, 8\}$. For $M = 1, 2, 4$, we set the action-sample batch size to $B = 8$, and for $M = 8$, we set $B = 16$. For the main comparison, we used Adam's default $\varepsilon = 10^{-8}$. We swept the learning rate over $\{10^{-4}, 2 \times 10^{-4}, 3 \times 10^{-4}, 5 \times 10^{-4}, 10^{-3}\}$ with 3 random seeds and selected a value that performed consistently well across environments and $M$ for fixed $\varepsilon = 10^{-8}$. As discussed in Sec. 5.3, a small value of Adam's $\varepsilon$ can mitigate the gradient-damping effect by normalizing gradient magnitudes. Thus, the main results with the default $\varepsilon = 10^{-8}$ primarily examined whether the directional entropy-increase effect of ReMax appears in practical actor–critic training. To study the role of Adam's $\varepsilon$ in the damping effect, we additionally varied $\varepsilon \in \{10^{-8}, 10^{-2}, 10^{-1}, 1.0\}$ for all $M$; the results are provided in App. C.2. We also tested the sensitivity of the results to the action-sample batch size $B \in \{8, 16\}$, with results reported in App. C.2.

**Main results.** Fig. 3 reports the average return with Adam's default $\varepsilon = 10^{-8}$. ReMAC with $M > 1$ achieved returns comparable to those of SAC and generally performed better than PPO in these experiments. In several environments, including Ant, Swimmer, and Walker2d, larger retry budgets ($M = 4, 8$) achieved higher average return than $M = 1$.

Fig. 4 reports the average policy entropy for different $M$. ReMAC with $M > 1$ yielded higher entropy than $M = 1$, and entropy generally increased with $M$, most clearly in Ant and Swimmer. This behavior is consistent with the qualitative trend observed in the toy analysis of Sec. 5.1. In some environments, such as HalfCheetah, larger $M$ led to lower entropy late in training; for example, $M = 8$ became lower than $M = 4$

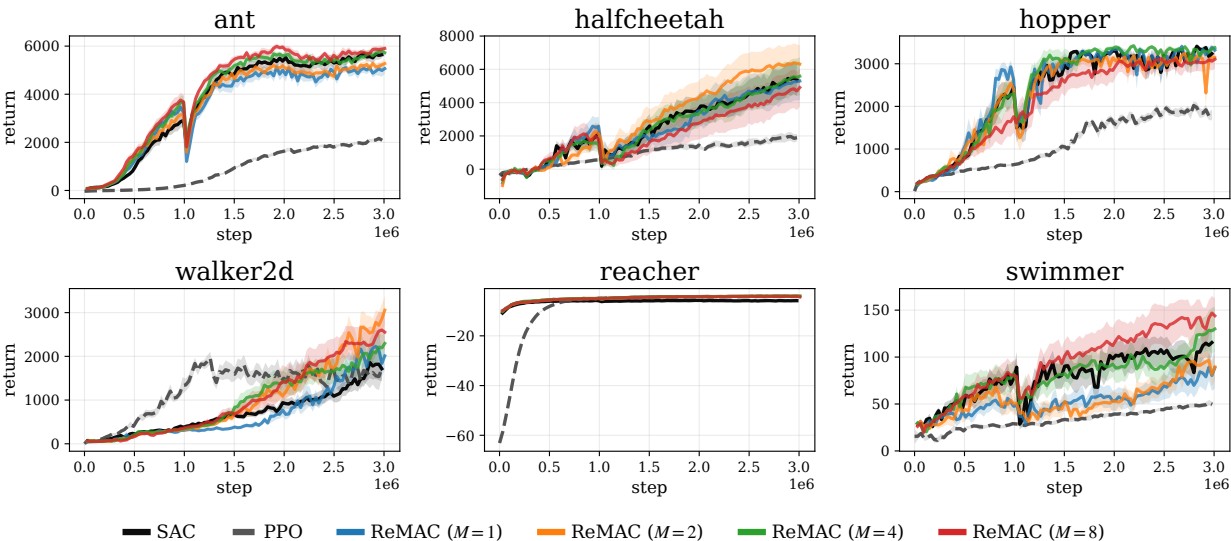

Figure 3: Average return of SAC, PPO, and ReMAC with different $M$ under Adam's default $\varepsilon = 10^{-8}$. The shaded area shows the standard error across 10 random seeds. ReMAC with $M > 1$ achieves performance comparable to SAC.

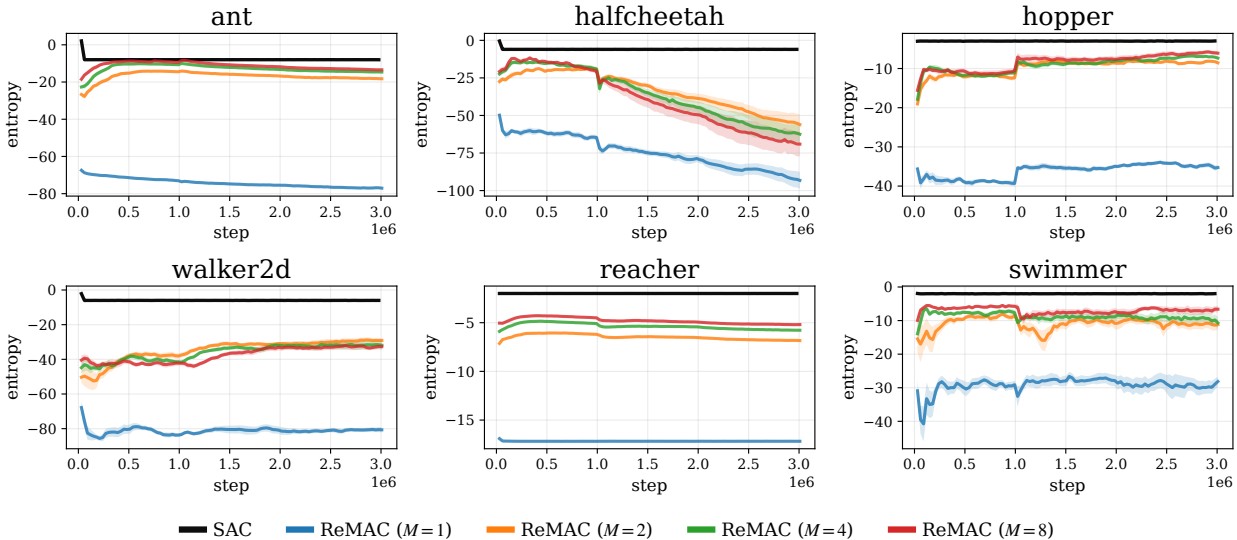

Figure 4: Average policy entropy of SAC and ReMAC with different $M$ under Adam's default $\varepsilon = 10^{-8}$. The shaded area shows the standard error across 10 random seeds. $M > 1$ generally yields higher entropy than $M = 1$, and entropy tends to increase with $M$ (most clearly in Ant).

and $M = 2$. We hypothesized that this behavior was caused by Adam's normalization counteracting the damping effect of ReMax, as discussed in Sec. 5.3. A more detailed analysis of the effect of $\varepsilon$ is provided in App. C.2, where increasing $\varepsilon$ tended to yield higher entropy, consistent with the damping interpretation. Finally, although ReMAC with $M > 1$ increased entropy relative to $M = 1$, its entropy remained lower than SAC, as expected because SAC's critic incorporated future entropy (Haarnoja et al., 2018a).

On HumanoidStandup ($d = 17$), ReMAC with $M \geq 2$ attained higher mean returns than SAC (App. C.3). From a practical perspective, these environment-step comparisons should be interpreted together with the

additional computational cost of ReMAC, which arose from the extra Q-network evaluations for $B$ action samples per state; see App. C for details.

**Choosing the retry budget.** No single $M$ is best across our environments, and performance is not monotone in $M$; nonetheless, $M > 1$ generally yields higher entropy than $M = 1$ and tends to match or improve on its return. Across the six main tasks, $M = 8$ yields higher final policy entropy than $M = 1$ at every tested $\varepsilon$ (App. C.2), although the ordering across intermediate values of $M$ is not monotone. We therefore treat $M$ as a hyperparameter that controls how strongly the objective favors stochasticity, much like the entropy coefficient of SAC—though, unlike an entropy bonus, it does not change the deterministic optimum of the idealized ReMax objective (Sec. 4.1). Since the cost of ReMAC grows with the action-sample batch size $B \geq M$, we recommend $M \in \{2, 4\}$ as practical default choices.

## 8 Conclusion

In this study, we extended the ReMax objective to continuous action spaces. We showed that ReMax reshapes the gradient landscape, guiding the policy toward higher entropy during optimization. We first demonstrated this effect via a toy vector-field analysis and then provided a theoretical explanation. On the practical side, we instantiated ReMax in an off-policy actor–critic algorithm, ReMax Actor-Critic (ReMAC), which can be implemented with minimal modifications to SAC. Empirically, we showed that ReMAC can promote higher policy entropy without entropy regularization and achieves performance comparable to SAC.

**Limitations and future work.** Our study has both theoretical and practical limitations. Theoretically, the entropy-increase result assumes smoothness and non-degenerate sample selection, whereas the damping bounds additionally assume strong convexity of the fixed-state cost $c(a) = -Q(s, a)$. These assumptions, particularly strong convexity, are not expected to hold for a general learned neural-network critic. Although the goal of our analysis is to build intuition, extending it to more general settings would help us better understand the properties of ReMax. Practically, our implementation is minimal and focuses on stochastic exploration, as discussed in Sec. 2; it could be improved by incorporating additional exploration mechanisms. Policy entropy, which our analysis and measurements concern, is only a proxy for exploration in the state space. Estimating the entropy of the visited-state distribution (App. C.6), we find no consistent improvement of ReMAC over SAC on these dense-reward locomotion tasks. Whether the higher entropy induced by ReMax translates into broader state-space exploration therefore remains open. Evaluating this question will require sparse-reward or hard-exploration domains, where temporally coherent or uncertainty-directed exploration may be important. A particularly well-aligned direction is to model the posterior over the Q-function (Osband et al., 2016; 2018), since ReMax explicitly considers return uncertainty. As suggested by Nishimori et al. (2026); Tong et al. (2026), this direction may enable deeper, structured exploration, which is important in sparse-reward settings. We hope that our analysis of ReMax's gradient properties and ReMAC's comparable performance to SAC with minimal modifications will provide a foundation for future work.

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

## Appendix Contents

## A Proofs of Propositions in Sec. 5

In this section, we prove the propositions in Sec. 5. Throughout this section, let

$$A_m = \mu + \sigma \xi_m, \qquad \xi_m \overset{\text{i.i.d.}}{\sim} \mathcal{N}(0, I_d).$$

**Lemma 1** (No ties from non-atomic cost values). Suppose Assumption 2 holds. Then, for every $\sigma \in (0, \bar{\sigma})$,

$$\mathbb{P}\left(\exists i \neq j \text{ such that } c(\mu + \sigma \xi_i) = c(\mu + \sigma \xi_j)\right) = 0.$$

**Proof.** Fix $\sigma \in (0, \bar{\sigma})$ and define $Y_i = c(\mu + \sigma \xi_i)$. Then $Y_1, \ldots, Y_M$ are independent and identically distributed. By Assumption 2, each $Y_i$ has no atoms. For any $i \neq j$,

$$\mathbb{P}(Y_i = Y_j) = \int_{\mathbb{R}} \mathbb{P}(Y_i = t) \, d\mathbb{P}_{Y_j}(t) = 0.$$

Taking a union bound over the finitely many pairs $(i, j)$ gives

$$\mathbb{P}(\exists i \neq j : Y_i = Y_j) = 0.$$

$\square$

**Lemma 2** (Differentiability of the scale objective). Assume Assumptions 1 and 2. For every $\sigma \in (0, \bar{\sigma})$, $J^M(\mu, \sigma)$ is differentiable with respect to $\sigma$ and

$$\partial_\sigma J^M(\mu, \sigma) = \mathbb{E}\left[-\nabla c(A_{m^\star})^\top \xi_{m^\star}\right],$$

where $m^\star \in \arg\min_{m=1,\ldots,M} c(A_m)$.

**Proof.** Fix $\sigma \in (0, \bar{\sigma})$. For a fixed realization of $\xi_{1:M}$, define

$$F(s) = \max_{m=1,\ldots,M} -c(\mu + s\xi_m).$$

By Lemma 1, the maximizer is almost surely unique at $s = \sigma$. Hence, by Danskin's theorem,

$$F'(\sigma) = -\nabla c(\mu + \sigma\xi_{m^\star})^\top \xi_{m^\star}$$

for almost every realization.

It remains to justify interchanging differentiation and expectation. Choose $\delta > 0$ such that $[\sigma - \delta, \sigma + \delta] \subset (0, \bar{\sigma})$. For every $s \in [\sigma - \delta, \sigma + \delta]$ and every $m$, Assumption 1 gives

$$\|\nabla c(\mu + s\xi_m)\| \leq \|\nabla c(\mu)\| + L(\sigma + \delta)\|\xi_m\|.$$

Therefore,

$$|F'(s)| \leq \left( \|\nabla c(\mu)\| + L(\sigma + \delta) \max_m \|\xi_m\| \right) \max_m \|\xi_m\|.$$

The right-hand side is integrable because $M < \infty$ and Gaussian random variables have finite moments. Dominated convergence yields

$$\partial_\sigma J^M(\mu, \sigma) = \mathbb{E}[F'(\sigma)] = \mathbb{E}\left[ -\nabla c(A_{m^\star})^\top \xi_{m^\star} \right].$$

$\square$

### A.1 Proof of Proposition 1

**Proof.** Let $g = \nabla c(\mu)$. By assumption, $g \neq 0$. Define $B_m = g^\top \xi_m$. Then $B_m \overset{d}{=} \|g\| Z_m$, where $Z_m \sim \mathcal{N}(0, 1)$. Since $g \neq 0$, the random variables $B_1, \ldots, B_M$ are continuous. Thus,

$$j^\star \in \arg \min_{m=1,\ldots,M} B_m$$

is almost surely unique.

We first show that, for almost every realization, $m^\star = j^\star$ for all sufficiently small $\sigma > 0$. By Assumption 1,

$$c(\mu + \sigma\xi_m) = c(\mu) + \sigma g^\top \xi_m + R_m(\sigma),$$

where

$$|R_m(\sigma)| \leq \frac{L}{2}\sigma^2 \|\xi_m\|^2.$$

Fix a realization for which $j^\star$ is unique, and define

$$\Delta = \min_{m \neq j^\star} (B_m - B_{j^\star}) > 0.$$

For any $m \neq j^\star$,

$$c(\mu + \sigma\xi_m) - c(\mu + \sigma\xi_{j^\star}) = \sigma(B_m - B_{j^\star}) + R_m(\sigma) - R_{j^\star}(\sigma).$$

Since

$$|R_m(\sigma) - R_{j^\star}(\sigma)| \leq \frac{L}{2}\sigma^2 \left( \|\xi_m\|^2 + \|\xi_{j^\star}\|^2 \right),$$

there exists $\sigma_1 > 0$ such that, for all $\sigma \in (0, \sigma_1)$,

$$|R_m(\sigma) - R_{j^\star}(\sigma)| \leq \frac{\sigma\Delta}{2} \qquad \text{for all } m \neq j^\star.$$

Thus,

$$c(\mu + \sigma\xi_m) - c(\mu + \sigma\xi_{j^\star}) \geq \frac{\sigma\Delta}{2} > 0$$

for all $m \neq j^\star$. Hence, $m^\star = j^\star$ for all sufficiently small $\sigma > 0$, almost surely.

By Lemma 2,

$$\partial_\sigma J^M(\mu, \sigma) = \mathbb{E}\left[-\nabla c(\mu + \sigma \xi_{m^\star})^\top \xi_{m^\star}\right].$$

Using the eventual equality $m^\star = j^\star$ and the continuity of $\nabla c$,

$$-\nabla c(\mu + \sigma \xi_{m^\star})^\top \xi_{m^\star} \to -g^\top \xi_{j^\star} = -\min_{m=1,\ldots,M} g^\top \xi_m$$

almost surely as $\sigma \downarrow 0$.

We now justify dominated convergence. For $0 < \sigma \leq 1$, Assumption 1 gives

$$\|\nabla c(\mu + \sigma \xi_{m^\star})\| \leq \|g\| + L \max_m \|\xi_m\|.$$

Therefore,

$$\left|\nabla c(\mu + \sigma \xi_{m^\star})^\top \xi_{m^\star}\right| \leq \left(\|g\| + L \max_m \|\xi_m\|\right) \max_m \|\xi_m\|.$$

The right-hand side is integrable. By dominated convergence,

$$\lim_{\sigma \downarrow 0} \partial_\sigma J^M(\mu, \sigma) = -\mathbb{E}\left[\min_{m=1,\ldots,M} g^\top \xi_m\right].$$

Since $g^\top \xi_m \stackrel{d}{=} \|g\| Z_m$,

$$-\mathbb{E}\left[\min_m g^\top \xi_m\right] = \|g\| \mathbb{E}\left[\max_m Z_m\right] = \|\nabla c(\mu)\| \beta_M,$$

where $\beta_M = \mathbb{E}[\max_{m=1,\ldots,M} Z_m]$. For $M \geq 2$, $\beta_M > 0$ because

$$\beta_M \geq \mathbb{E}[\max\{Z_1, Z_2\}] = \frac{1}{2}\mathbb{E}|Z_1 - Z_2| > 0.$$

Therefore,

$$\lim_{\sigma \downarrow 0} \partial_\sigma J^M(\mu, \sigma) = \|\nabla c(\mu)\| \beta_M > 0.$$

By the definition of a one-sided limit, there exists $\sigma_0 > 0$ such that

$$\partial_\sigma J^M(\mu, \sigma) > 0 \qquad \text{for all } \sigma \in (0, \sigma_0).$$

$\square$

## A.2 Proof of Proposition 2

**Proof.** For $M = 1$,

$$J^1(\mu, \sigma) = -\mathbb{E}[c(\mu + \sigma \xi)].$$

By Assumptions 1 and 5, differentiation under the expectation is justified by dominated convergence, and

$$\partial_\sigma J^1(\mu, \sigma) = -\mathbb{E}[\nabla c(\mu + \sigma \xi)^\top \xi].$$

Let $A = \mu + \sigma \xi$. By Gaussian integration by parts, for each coordinate $i$,

$$\mathbb{E}[\partial_i c(A) \xi_i] = \sigma \mathbb{E}[\partial_{ii}^2 c(A)].$$

Summing over $i = 1, \ldots, d$ yields

$$\mathbb{E}[\nabla c(A)^\top \xi] = \sigma \mathbb{E}[\text{tr}(\nabla^2 c(A))].$$

By Assumptions 4 and 5,

$$\nabla^2 c(A) \succeq \lambda I_d.$$

Therefore,

$$\text{tr}(\nabla^2 c(A)) \geq \lambda d.$$

Hence,

$$\partial_\sigma J^1(\mu, \sigma) = -\sigma \mathbb{E}[\text{tr}(\nabla^2 c(A))] \leq -\lambda d \sigma < 0.$$

$\square$

**Lemma 3** (No ties under strong convexity)**.** Assume Assumptions 1, 3, and 4. Let $X_1, \ldots, X_M$ be independent random variables on $\mathbb{R}^d$ with densities with respect to Lebesgue measure. Then

$$\mathbb{P}\left(\exists i \neq j \text{ such that } c(X_i) = c(X_j)\right) = 0.$$

*Proof.* We first show that each level set of $c$ has Lebesgue measure zero. By Assumptions 3 and 4, 0 is the unique minimizer of $c$, and $c(0) = 0$.

Fix $t \in \mathbb{R}$. If $t < 0$, then $\{x : c(x) = t\}$ is empty. If $t = 0$, then $\{x : c(x) = t\} = \{0\}$, which has Lebesgue measure zero. Now suppose $t > 0$ and define

$$C_t = \{x \in \mathbb{R}^d : c(x) \leq t\}.$$

The set $C_t$ is convex. It also has nonempty interior because Assumptions 1 and 3 imply $c(x) \leq (L/2)\|x\|^2$, so a sufficiently small ball around 0 is contained in $C_t$. We claim that

$$\{x : c(x) = t\} \subseteq \partial C_t.$$

Suppose, for contradiction, that $x \in \{c = t\}$ belongs to the interior of $C_t$. Then there exists a nonzero vector $u$ such that $x + u \in C_t$ and $x - u \in C_t$. By strong convexity,

$$c(x) = c\left(\frac{x+u+x-u}{2}\right) \leq \frac{c(x+u) + c(x-u)}{2} - \frac{\lambda}{2}\|u\|^2 \leq t - \frac{\lambda}{2}\|u\|^2 < t,$$

contradicting $c(x) = t$. Thus, the level set is contained in the boundary of the convex set $C_t$. The boundary of a convex set in $\mathbb{R}^d$ has Lebesgue measure zero. Hence every level set of $c$ has Lebesgue measure zero.

Since each $X_i$ has a density,

$$\mathbb{P}(c(X_i) = t) = \mathbb{P}(X_i \in \{x : c(x) = t\}) = 0$$

for every $t \in \mathbb{R}$. Thus $c(X_i)$ has no atoms. The same argument as in Lemma 1 gives

$$\mathbb{P}(\exists i \neq j : c(X_i) = c(X_j)) = 0.$$

$\square$

### A.3 Proof of Proposition 3

*Proof.* Assumptions 3 and 4 imply

$$c(a) \geq \frac{\lambda}{2}\|a\|^2 \qquad \text{for all } a \in \mathbb{R}^d.$$

Assumptions 1 and 3 imply

$$c(a) \leq \frac{L}{2}\|a\|^2 \qquad \text{for all } a \in \mathbb{R}^d,$$

and

$$\|\nabla c(a)\| = \|\nabla c(a) - \nabla c(0)\| \leq L\|a\|.$$

By Lemma 3, the selected index

$$m^\star \in \arg \min_{m=1,\ldots,M} c(A_m)$$

is almost surely unique. For a fixed realization of $\xi_{1:M}$, define

$$F(\mu, \sigma) = \max_{m=1,\ldots,M} -c(\mu + \sigma \xi_m).$$

By Danskin's theorem,

$$\nabla_\mu F(\mu, \sigma) = -\nabla c(A_{m^\star}),$$

and

$$\partial_\sigma F(\mu, \sigma) = -\nabla c(A_{m^\star})^\top \xi_{m^\star}$$

almost surely. Furthermore,

$$\|\nabla c(A_{m^\star})\| \le L\|A_{m^\star}\| \le L\left(\|\mu\| + \sigma \max_m \|\xi_m\|\right),$$

and

$$|\nabla c(A_{m^\star})^\top \xi_{m^\star}| \le L\|A_{m^\star}\|\|\xi_{m^\star}\| \le L\left(\|\mu\| + \sigma \max_m \|\xi_m\|\right) \max_m \|\xi_m\|.$$

These bounds hold uniformly on a neighborhood of the evaluation point: for every $(\mu', \sigma')$ with $\|\mu' - \mu\| \le \delta$ and $|\sigma' - \sigma| \le \delta$, both quantities are dominated by

$$L\left(\|\mu\| + \delta + (\sigma + \delta)\max_m \|\xi_m\|\right)\left(1 + \max_m \|\xi_m\|\right),$$

which is integrable because $M < \infty$ and Gaussian moments are finite. Thus, differentiation can be interchanged with expectation by dominated convergence, yielding

$$\nabla_\mu J^M(\mu, \sigma) = \mathbb{E}[-\nabla c(A_{m^\star})],$$

and

$$\partial_\sigma J^M(\mu, \sigma) = \mathbb{E}[-\nabla c(A_{m^\star})^\top \xi_{m^\star}].$$

Using $\|\nabla c(a)\| \le L\|a\|$,

$$\|\nabla_\mu J^M(\mu, \sigma)\| \le \mathbb{E}[\|\nabla c(A_{m^\star})\|] \le L\mathbb{E}[\|A_{m^\star}\|].$$

Similarly,

$$|\partial_\sigma J^M(\mu, \sigma)| \le \mathbb{E}[\|\nabla c(A_{m^\star})\|\|\xi_{m^\star}\|] \le L\mathbb{E}[\|A_{m^\star}\|\|\xi_{m^\star}\|].$$

Since $m^\star$ minimizes $c(A_m)$,

$$c(A_{m^\star}) = \min_{m=1,\dots,M} c(A_m).$$

Using the quadratic bounds on $c$,

$$\frac{\lambda}{2}\|A_{m^\star}\|^2 \le c(A_{m^\star}) = \min_m c(A_m) \le \frac{L}{2}\min_m \|A_m\|^2.$$

Therefore,

$$\|A_{m^\star}\| \le \sqrt{\frac{L}{\lambda}} \min_{m=1,\dots,M} \|A_m\|.$$

Substituting this bound gives

$$\|\nabla_\mu J^M(\mu, \sigma)\| \le L\sqrt{\frac{L}{\lambda}} \mathbb{E}\left[\min_m \|A_m\|\right], \qquad |\partial_\sigma J^M(\mu, \sigma)| \le L\sqrt{\frac{L}{\lambda}} \mathbb{E}\left[\min_m \|A_m\|\|\xi_{m^\star}\|\right]. \tag{5}$$

$\square$

## A.4  Proof of Corollary 1

***Proof.*** Recall $\kappa_{d,M} = \mathbb{E}[\min_{m \le M} \|\xi_m\|]$, $\tau_{d,M} = \mathbb{E}[\min_{m \le M} \|\xi_m\|^2]$, and $\nu_{d,M} = (\mathbb{E}[\max_{m \le M} \|\xi_m\|^2])^{1/2}$. We bound the two expectations of Eq. (5) in turn.

*Step 1: the mean gradient.* For every $m$, the triangle inequality gives $\|A_m\| \le \|\mu\| + \sigma\|\xi_m\|$. Evaluating the right-hand side at an index attaining $\min_m \|\xi_m\|$, and using that a minimum is no larger than the value at any particular index,

$$\min_{m \le M} \|A_m\| \le \|\mu\| + \sigma \min_{m \le M} \|\xi_m\| \qquad \text{almost surely.} \tag{6}$$

Taking expectations, $\mathbb{E}[\min_m \|A_m\|] \leq \|\mu\| + \sigma\kappa_{d,M}$, which with Eq. (5) gives the first bound.

*Step 2: the scale gradient.* Since $m^\star$ is one of the $M$ indices, $\|\xi_{m^\star}\| \leq \max_{m \leq M} \|\xi_m\|$ almost surely, so by Cauchy–Schwarz

$$\mathbb{E}\left[\min_m \|A_m\| \, \|\xi_{m^\star}\|\right] \leq \left(\mathbb{E}\left[\left(\min_m \|A_m\|\right)^2\right]\right)^{1/2} \left(\mathbb{E}\left[\max_m \|\xi_m\|^2\right]\right)^{1/2} = \left(\mathbb{E}\left[\left(\min_m \|A_m\|\right)^2\right]\right)^{1/2} \nu_{d,M}.$$

Squaring Eq. (6) and taking expectations,

$$\mathbb{E}\left[\left(\min_m \|A_m\|\right)^2\right] \leq \|\mu\|^2 + 2\|\mu\|\sigma\kappa_{d,M} + \sigma^2\tau_{d,M} \leq \left(\|\mu\| + \sigma\sqrt{\tau_{d,M}}\right)^2,$$

where the last step uses $\kappa_{d,M} \leq \sqrt{\tau_{d,M}}$, which is Jensen's inequality applied to $x \mapsto x^2$. Combining the two displays gives the second bound.

*Step 3: monotonicity in $M$.* Write $X_M = \min_{m \leq M} \|\xi_m\|$. Then $X_{M+1} = \min\{X_M, \|\xi_{M+1}\|\} \leq X_M$ almost surely, and $\mathbb{P}(\|\xi_{M+1}\| < X_M) > 0$ because $\|\xi\|$ has full support on $(0, \infty)$. Hence $\mathbb{E}[X_{M+1}] < \mathbb{E}[X_M]$, so $\kappa_{d,M}$ is strictly decreasing in $M$; the same argument applied to $X_M^2$ gives strict monotonicity of $\tau_{d,M}$.

*Step 4: the rate.* Let $\chi_m = \|\xi_m\|$, which is chi-distributed with $d$ degrees of freedom, with distribution function $F$ and density $f(t) = t^{d-1}e^{-t^2/2}/(2^{d/2-1}\Gamma(d/2))$. For $t \in [0, 1]$ we have $e^{-u^2/2} \geq e^{-1/2}$ on $[0, t]$, so

$$F(t) \geq \frac{e^{-1/2}}{2^{d/2-1}\Gamma(d/2)} \int_0^t u^{d-1} \, du = \frac{e^{-1/2}}{2^{d/2}\,\Gamma(\frac{d}{2}+1)} t^d = c_d \, t^d,$$

using $\Gamma(d/2+1) = (d/2)\Gamma(d/2)$; in particular $F(1) \geq c_d$. By the layer-cake representation and independence, $\mathbb{P}(X_M > t) = (1 - F(t))^M$, so $\kappa_{d,M} = \int_0^1 (1 - F)^M dt + \int_1^\infty (1 - F)^M dt$. For the first integral, $1 - x \leq e^{-x}$ and $F(t) \geq c_d t^d$ on $[0, 1]$ give

$$\int_0^1 (1 - F(t))^M dt \leq \int_0^\infty e^{-Mc_d t^d} dt = \Gamma\left(1 + \tfrac{1}{d}\right)(c_d M)^{-1/d},$$

by the substitution $u = Mc_d t^d$. For the second, $1 - F(t) \leq 1 - F(1) \leq 1 - c_d$ for $t \geq 1$, hence

$$\int_1^\infty (1 - F(t))^M dt \leq (1 - c_d)^{M-1} \int_0^\infty \mathbb{P}(\chi_1 > t) \, dt = (1 - c_d)^{M-1} \mathbb{E}[\chi_1] \leq \sqrt{d}\,(1 - c_d)^{M-1},$$

using $\mathbb{E}[\chi_1] \leq (\mathbb{E}[\chi_1^2])^{1/2} = \sqrt{d}$. Adding the two gives the stated bound on $\kappa_{d,M}$. Finally, $(\xi_1, \ldots, \xi_M) \mapsto \max_{m \leq M} \|\xi_m\|$ is 1-Lipschitz on $\mathbb{R}^{dM}$, so Gaussian concentration (Boucheron et al., 2013) gives $\mathbb{E}[\max_m \|\xi_m\|] \leq \sqrt{d} + \sqrt{2\log M}$, and the Gaussian Poincaré inequality bounds its variance by 1. Therefore $\nu_{d,M}^2 = \mathrm{Var}(\max_m \|\xi_m\|) + (\mathbb{E}[\max_m \|\xi_m\|])^2 \leq 1 + (\sqrt{d} + \sqrt{2\log M})^2$. $\qquad\square$

# B    Details of the gradient analysis

We provide additional details for the vector field analysis in Sec. 5.1 and the effect of Adam's normalization in Sec. 5.3.

## B.1    Details for the vector field analysis

In Sec. 5.1, we visualized the gradient landscape of the ReMax objective to build intuition for its exploration-encouraging properties. Here we describe the setup and methodology used to generate these vector fields.

We consider a stateless continuous-control problem in which the policy is a one-dimensional Gaussian distribution $\mathcal{N}(\mu, \sigma^2)$. To enable differentiation through sampling, actions are parameterized using the reparameterization trick, $a_i = \mu + \sigma\xi_i$, where $\xi_i \sim \mathcal{N}(0, 1)$ are i.i.d. standard normal noise variables. We use

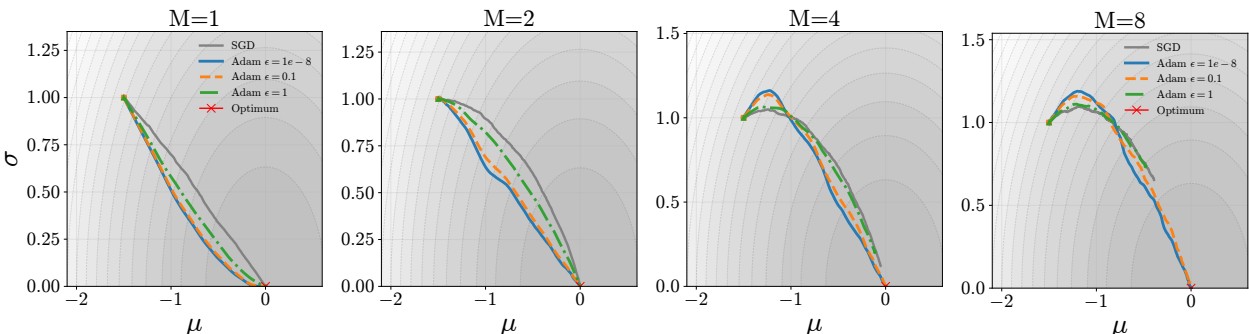

Figure 5: The two-dimensional trajectories of the parameter $(\mu, \sigma)$ optimized by Adam starting from $(\mu, \sigma) = (-1.5, 1.0)$ with different $\varepsilon$ values ($\varepsilon \in \{10^{-8}, 10^{-1}, 1.0\}$) and by SGD. As the vector field indicates, increasing $M$ makes the path deviate more from the optimum and slows convergence. However, with smaller $\varepsilon$, convergence near the optimum remains fast, whereas larger $\varepsilon$ aligns the trajectory with the SGD path that follows the vector field.

the quadratic deterministic reward $r(a) = -a^2$, making $a = 0$ the unique optimal action. For entropy regularization, we set $\alpha = 0.5$ and use the policy-level objective $J_{\text{ent}}(\mu, \sigma) = \mathbb{E}_{a \sim \mathcal{N}(\mu, \sigma^2)}[r(a)] + \alpha \mathcal{H}(\mathcal{N}(\mu, \sigma^2))$, where $\mathcal{H}(\mathcal{N}(\mu, \sigma^2)) = \frac{1}{2} \log(2\pi e \sigma^2)$.

To construct the vector field, we evaluate gradients of the ReMax objective with respect to $\mu$ and $\sigma$ on a grid defined by $\mu \in [-2, 2]$ and $\sigma \in [0.1, 2]$. For each coordinate $(\mu, \sigma)$, we sample a batch of $B = 16$ actions and compute gradients using the reparameterization (RP) estimator with automatic differentiation. A single stochastic gradient evaluation would produce a noisy vector field. To approximate the expected gradient landscape $\mathbb{E}[\nabla_{\mu, \sigma} \mathcal{J}^M]$, we therefore average RP gradients over 100 independent Monte Carlo trials for each grid coordinate.

### B.2 Details for Adam's normalization

In Sec. 5.3, we showed that Adam's normalization can mitigate the damping effect of ReMax. To illustrate this, we plot the convergence path of $(\mu, \sigma)$ in the two-dimensional parameter space in Fig. 5. As $M$ increases, the trajectory becomes more gradual around the optimum, consistent with the vector-field analysis.

Furthermore, the consistency with SGD observed for larger $\varepsilon$ in Fig. 2 also appears in these trajectories, supporting the argument that increasing $\varepsilon$ makes the optimization path follow the vector field more closely.

**Isolating the denominator.** Changing $\varepsilon$ rescales the adaptive normalization but never removes it. To isolate the denominator we therefore compare an *EMA-only* variant, $\theta_{t+1} = \theta_t - \alpha \hat{m}_t$, which removes the entire adaptive second-moment denominator while retaining Adam's first-moment EMA and its bias correction. (Because the denominator is deleted, the second-moment estimate $v_t$, the constant $\beta_2$, and the second-moment bias correction play no role in this variant.) Fig. 6 compares it with Adam at $\varepsilon \in \{10^{-8}, 10^{-1}, 1\}$ and with SGD on the same toy problem ($\alpha = 0.01$, $B = 64$, $T = 1000$, 5 seeds), and Tab. 1 quantifies the resulting slowdown. EMA-only retains the full $M$-dependent slowdown and reproduces SGD's behavior almost exactly, whereas Adam with $\varepsilon = 10^{-8}$ removes nearly all of it. The first-moment bias correction is retained in EMA-only, yet the full $M$-dependent slowdown remains. This supports the interpretation that the adaptive second-moment denominator, rather than the first-moment bias correction, is responsible for removing much of the gradient-magnitude effect.

## C Details of the experiments

Here we provide additional details for the experiments in Sec. 7.

Table 1: Isolating Adam's denominator on the toy problem. Slowdown factor at each $M$: the final distance to the optimum relative to the same optimizer at $M = 1$ (mean over 5 seeds), so a larger value means more of the $M$-dependent damping survives. Deleting the denominator (EMA-only) preserves the damping and matches SGD; a small $\varepsilon$ removes it.

|  | $M = 2$ | $M = 4$ | $M = 8$ |
|---|---|---|---|
| Adam ($\varepsilon = 10^{-8}$) | 1.0 | 1.0 | 1.9 |
| Adam ($\varepsilon = 10^{-1}$) | 1.0 | 1.1 | 418 |
| Adam ($\varepsilon = 1$) | 69 | 1987 | 7426 |
| EMA-only (no denominator) | 5.8 | 1290 | 7584 |
| SGD | 10 | 1358 | 7614 |

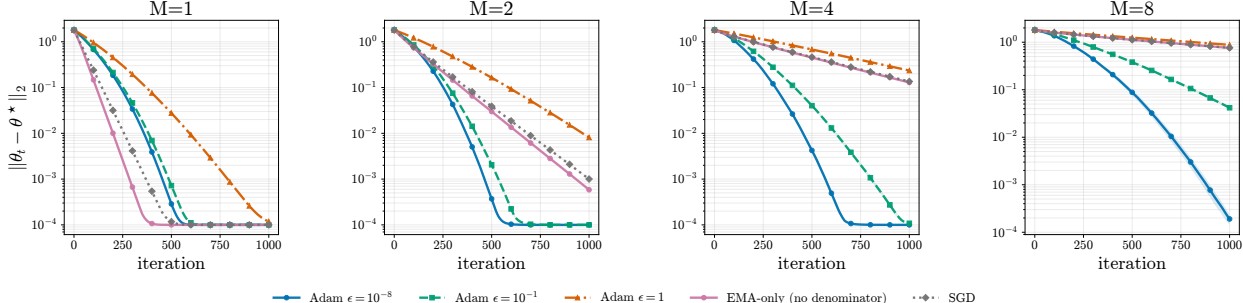

Figure 6: Isolating Adam's denominator. Distance to the optimum $(\mu, \sigma) = (0, 0)$ for Adam ($\varepsilon \in \{10^{-8}, 10^{-1}, 1\}$), for EMA-only (Adam with the denominator removed, $\theta_{t+1} = \theta_t - \alpha \hat{m}_t$), and for SGD; shading is the standard error over 5 seeds. EMA-only preserves the $M$-dependent slowdown and coincides with SGD and with large-$\varepsilon$ Adam, whereas $\varepsilon = 10^{-8}$ largely removes it.

## C.1 Setup

**Implementation.** The implementation of the ReMAC algorithm is based on the `rejax`[1] implementation of SAC. We include the code in the supplementary material.

**Statistics.** Unless stated otherwise, the *final* value of a run is the mean over the last 10% of training, which is robust to evaluation noise, and we report the mean $\pm$ the standard deviation across the 10 seeds.

**Hyperparameters.** We follow the SAC and PPO implementations and hyperparameters provided in the `rejax` library (Liesen et al., 2024), which contains environment-specific tuned configurations for Brax tasks. Listing all hyperparameters would largely duplicate that repository, so we refer readers to the environment-specific configuration files in the supplementary material at `brax/configs/` for the exact configurations used in our experiments. The hyperparameters (learning rate, batch size, discount factor, Polyak factor, etc.) were tuned for each environment by the authors of `rejax`. We tuned the learning rate for ReMAC by sweeping $\{10^{-4}, 2 \times 10^{-4}, 3 \times 10^{-4}, 5 \times 10^{-4}, 10^{-3}\}$ with 3 random seeds and selecting a value that performed reasonably well across environments and $M$ for fixed $\varepsilon = 10^{-8}$. The selected learning rates are provided in Tab. 2. The other hyperparameters are the same as those used for SAC and PPO in `rejax`.

## C.2 Additional results

Here we provide additional results for the experiments in Sec. 7.

**Entropy and return with different** $\varepsilon$ We plot policy entropy and return for each $\varepsilon \in \{10^{-8}, 10^{-2}, 10^{-1}, 1\}$ in Figs. 7–14, with $M \in \{1, 2, 4, 8\}$ overlaid in each panel. This organization makes

---

[1]https://github.com/keraJLi/rejax

Table 2: Selected hyperparameters for ReMAC.

| Environment | Learning Rate |
|---|---|
| HalfCheetah | $10^{-4}$ |
| Ant | $2 \times 10^{-4}$ |
| Hopper | $3 \times 10^{-4}$ |
| Reacher | $3 \times 10^{-4}$ |
| Swimmer | $10^{-4}$ |
| Walker2d | $10^{-4}$ |
| HumanoidStandup | $10^{-4}$ |

the effect of varying the retry budget directly visible at a fixed optimizer setting. Within a panel, $M$ carries the color that Figs. 3 and 4 assign to it, together with a distinct line style and marker. For $M = 1$, entropy increases with $\varepsilon$ in every task. For $M > 1$, larger $\varepsilon$ generally yields higher entropy, although the trend is not monotone in every environment. This is consistent with Sec. 5.3, while the return tends to decrease for larger $\varepsilon$, because a larger $\varepsilon$ reduces the effective step size and can slow learning. Comparing across the figures, the final policy entropy at $M = 8$ exceeds that at $M = 1$ in all six environments at every $\varepsilon$, and across these settings $M = 1$ exhibits the strongest and most consistent entropy reduction; the ordering in $M$ is not strictly monotone in every environment.

**Ablation by the Learning Rate.** Since Adam's $\varepsilon$ rescales the per-parameter step size, a natural question is whether its effect can be reproduced simply by changing the global learning rate $\alpha$. To test this, we fix $M = 4$, sweep $\alpha$ for both the default $\varepsilon = 10^{-8}$ and the large value $\varepsilon = 1$, and plot the resulting return and entropy in Figs. 15 and 16. We selected three environments (HalfCheetah, Swimmer, and Walker2d) because their default learning rate is $10^{-4}$, which makes it easy to compare the learning-rate effect. At $\varepsilon = 1$, where the large denominator strongly damps the effective step size, increasing $\alpha$ from $10^{-4}$ to $10^{-3}$ restores the effective step size and recovers most of the return lost relative to $\varepsilon = 10^{-8}$, while the entropy decreases as $\alpha$ grows yet stays above the $\varepsilon = 10^{-8}$ level, especially in HalfCheetah and Swimmer. At $\varepsilon = 10^{-8}$, the learning rate tuned for SAC ($\alpha = 10^{-4}$) already attains the highest return, and decreasing $\alpha$ from $10^{-4}$ to $10^{-5}$ only slows learning while the entropy still collapses to the low values characteristic of small $\varepsilon$. Hence $\alpha$ and $\varepsilon$ are not interchangeable: $\alpha$ scales the whole update uniformly, whereas $\varepsilon$ reshapes the per-coordinate normalization and thereby controls the entropy, consistent with the analysis in Sec. 5.3. Furthermore, we found that the best average return is achieved either with $\varepsilon = 1$ (HalfCheetah) or with $\varepsilon = 10^{-8}$ (Walker2d and Swimmer), depending on the environment, further suggesting the importance of tuning $\varepsilon$ and $\alpha$ jointly to achieve the best performance with ReMAC.

**Ablation by the Batch Size.** The unbiased ReMax estimator draws $B$ action samples per state to estimate the order-$M$ objective and is only defined for $B \geq M$, with larger $B$ reducing the variance of the estimate. We ablate $B \in \{8, 16\}$ at $\varepsilon = 10^{-8}$ with the tuned learning rate for $M \in \{1, 2, 4, 8\}$ in Fig. 17. Across all tasks and all $M$, the standard-error bands substantially overlap: $B = 16$ is comparable to, and occasionally marginally better than, $B = 8$, and the relative ordering across $M$ is unchanged. Because the estimator becomes noisier as $M$ approaches $B$, we use $B = 16$ for the largest setting $M = 8$ and $B = 8$ otherwise; this ablation confirms that the choice of $B$ does not materially affect the reported results.

**Computational cost.** We report the wall-clock training time of SAC and ReMAC with $B = 8$ and $B = 16$ on HalfCheetah for 3M environment steps, vectorized over 10 random seeds. This setting provides a controlled comparison because SAC and ReMAC share the same training pipeline and network architectures and differ mainly in the actor loss, while ReMAC also omits the temperature update used for entropy regularization. The evaluation frequency is once every 30,000 environment steps, and all experiments are run on a GeForce RTX 2080 Ti. The results are reported in Tab. 3. ReMAC is slower than SAC because its actor update requires additional Q-network evaluations for $B$ action samples per state. As expected, the wall-clock time increases with $B$. This additional computational cost is a limitation of the current implementation.

Table 3: Wall-clock training time of SAC and ReMAC with $B = 8$ and $B = 16$ on HalfCheetah for 3M environment steps, vectorized over 10 random seeds.

| SAC | ReMAC ($B = 8$) | ReMAC ($B = 16$) |
|---|---|---|
| 1 h 03 min | 1 h 33 min | 2 h 06 min |

### C.3 HumanoidStandup

The six tasks in Sec. 7 have at most 8 action dimensions, so we additionally evaluated on HumanoidStandup, which has $d = 17$ and is the highest-dimensional environment for which `rejax` provides a tuned configuration (the library does not ship a Humanoid configuration). Tab. 4 reports the results. The average return increases across the tested retry budgets, and ReMAC with $M \geq 2$ attains a higher average return than SAC and $M = 1$. Policy entropy is far below SAC's on this task and is not ordered by $M$, in contrast to the lower-dimensional environments.

Table 4: HumanoidStandup ($d = 17$), Adam's default $\varepsilon = 10^{-8}$. Final return and policy entropy, mean $\pm$ s.d. over $n = 10$ seeds (App. C.1). ReMAC with $M \geq 2$ attains a higher average return than SAC. Policy entropy, unlike return, is not ordered by $M$.

|  | Return | Policy entropy |
|---|---|---|
| PPO | $12312 \pm 634$ | — |
| SAC | $25427 \pm 1081$ | $-17 \pm 0$ |
| ReMAC ($M = 1$) | $25209 \pm 1138$ | $-324 \pm 30$ |
| ReMAC ($M = 2$) | $26049 \pm 808$ | $-249 \pm 34$ |
| ReMAC ($M = 4$) | $27003 \pm 761$ | $-347 \pm 25$ |
| ReMAC ($M = 8$) | $27575 \pm 497$ | $-315 \pm 19$ |

**Effect of Adam's $\varepsilon$.** At the default $\varepsilon = 10^{-8}$, a larger retry budget does not reliably raise policy entropy on this task: $M = 4$ and $M = 8$ are close to or below $M = 1$ (Tab. 4), consistent with Adam's normalization counteracting the ReMax entropy-increase effect (Sec. 5.3). As $\varepsilon$ increases this normalization weakens, and the entropy-increase effect of $M$ becomes visible: at $\varepsilon = 1$ every $M > 1$ exceeds $M = 1$, and overall entropy rises with $\varepsilon$ at every $M$ (Tab. 5). Return does not degrade over this range and improves slightly for $M \geq 2$ (e.g. $M = 8$: $27575 \pm 497 \to 28368 \pm 484$ from $\varepsilon = 10^{-8}$ to $\varepsilon = 1$).

Table 5: HumanoidStandup policy entropy and return across Adam's $\varepsilon$, mean $\pm$ s.d. over $n = 10$ seeds. Larger $\varepsilon$ raises entropy at every $M$ and makes the entropy-increase effect of $M > 1$ visible (at $\varepsilon = 1$ every $M > 1$ exceeds $M = 1$), while return does not degrade and improves slightly for $M \geq 2$.

|  | Policy entropy | | | Return | | |
|---|---|---|---|---|---|---|
|  | $\varepsilon=10^{-8}$ | $\varepsilon=10^{-1}$ | $\varepsilon=1$ | $\varepsilon=10^{-8}$ | $\varepsilon=10^{-1}$ | $\varepsilon=1$ |
| ReMAC ($M$=1) | $-324 \pm 30$ | $-198 \pm 17$ | $-128 \pm 11$ | $25209 \pm 1138$ | $25342 \pm 1024$ | $25066 \pm 1792$ |
| ReMAC ($M$=2) | $-249 \pm 34$ | $-146 \pm 23$ | $-70 \pm 9$ | $26049 \pm 808$ | $26471 \pm 1118$ | $26977 \pm 792$ |
| ReMAC ($M$=4) | $-347 \pm 25$ | $-177 \pm 18$ | $-79 \pm 12$ | $27003 \pm 761$ | $27360 \pm 920$ | $27719 \pm 796$ |
| ReMAC ($M$=8) | $-315 \pm 19$ | $-200 \pm 12$ | $-87 \pm 6$ | $27575 \pm 497$ | $27843 \pm 596$ | $28368 \pm 484$ |

### C.4 Do the analyzed mechanisms have measurable analogues on Brax?

Sec. 5 identifies two effects in the fixed-state, fixed-critic setting: a positive scale gradient for $M > 1$ at sufficiently small scale, and best-sample gradient bounds that vanish at the deterministic optimum. For the mean gradient, the bound provides an $M$-decreasing upper envelope. Both have measurable analogues during training, so we instrumented ReMAC to log them on {HalfCheetah, Ant, Hopper} $\times M \in \{1, 2, 4\}$ with 10 seeds, for 3M steps under the tuned learning rates. Final values follow the convention of App. C.1. The runs for different $M$ are trained separately and therefore reach different actor and critic parameters,

replay and state distributions, and optimizer states. The comparison below is consequently not a controlled isolation of the retry weighting, and we read it as evidence consistent with the proposed mechanism rather than as a causal measurement of it.

**Gradient magnitude: the gradient, and the Adam update.** Fig. 18 plots the actor gradient norm and the realized update norm, and Tab. 6 reports the ratio between $M = 4$ and $M = 1$ for both. The gradient norm decreases with $M$ in every environment and under both optimizers: at $M = 4$ it is reduced to approximately 17–75% of the corresponding $M = 1$ value, depending on the environment and the optimizer. This trend is consistent with the $M$-decreasing upper envelope of Corollary 1. The update norm behaves differently, in the direction anticipated by Sec. 5.3. Under SGD, where the update is proportional to the gradient, the update norm follows the gradient norm. Under Adam with $\varepsilon = 10^{-8}$ it does not: the ratios stay at or slightly above 1. This is consistent with per-coordinate normalization by $\sqrt{\hat{v}_t}$ dividing out the scale of the gradient, so that the reduction visible in the gradient does not reach the update.

Table 6: Gradient damping on Brax. Ratio of the actor gradient norm $\|\nabla_\theta L_{\text{actor}}^M\|$, and of the realized update norm $\|\Delta\theta\|$, at $M = 4$ relative to $M = 1$ (final values, $n = 10$ seeds). A ratio below 1 means the quantity is smaller at $M = 4$ than at $M = 1$. The gradient norm is reduced under both optimizers; the update norm is reduced under SGD but not under Adam, whose normalization divides out the scale of the gradient.

| Actor optimizer | | HalfCheetah | Ant | Hopper |
|---|---|---|---|---|
| Adam | $\|\nabla_\theta L_{\text{actor}}^M\|$ | 0.45 | 0.75 | 0.43 |
| Adam | $\|\Delta\theta\|$ | 1.10 | 1.00 | 1.07 |
| SGD | $\|\nabla_\theta L_{\text{actor}}^M\|$ | 0.46 | 0.65 | 0.17 |
| SGD | $\|\Delta\theta\|$ | 0.46 | 0.65 | 0.17 |

**The scale gradient.** The actor network emits the pre-squash log-scale as $\log \sigma_\theta(s) = W_\sigma h_\theta(s) + b_\sigma$, an affine head on its shared features $h_\theta(s)$. Because the bias $b_\sigma \in \mathbb{R}^d$ shifts $\log \sigma_\theta(s)$ by the same amount at every state, the gradient of the actor loss with respect to $b_\sigma$ is the minibatch average of the per-state log-scale gradients; we call its mean over the $d$ action coordinates the *scale gradient*. The loss is the negative objective, so a *negative* scale gradient means the update pushes the policy scale *up*.

Tab. 7 reports how often the scale gradient is negative, together with the final policy scale. Both increase with $M$ in all three environments: a larger retry budget spends more of the run pushing the scale up, and ends at a scale several hundred to more than ten thousand times larger than at $M = 1$. This is the direction of the mechanism in Proposition 1, though not a direct verification of it: the differences in the fraction are small relative to the spread across seeds, and the scale gradient is a network-level proxy rather than the scalar $\partial_\sigma J^M$ of the analysis, since the implemented policy is state-dependent, diagonal, squashed by a tanh, and parameterized by a neural network. We summarize the scale gradient by this fraction rather than by its run mean because the runs separate in scale before the first logged evaluation (Fig. 4), which leaves the mean dominated by the long phase in which each $M$ already sits at its own scale.

Table 7: The scale gradient on Brax, with an Adam actor. Left: the fraction of logged updates whose scale gradient is negative, i.e. that push the pre-squash scale $\sigma$ up (mean $\pm$ s.d. over $n = 10$ seeds). We summarize the scale gradient this way rather than by its run mean, which is dominated by the phase after the runs have already separated in $\sigma$ (see text). This is a network-level proxy for the scale-gradient effect, not the scalar $\partial_\sigma J^M$ of the analysis. Right: the resulting pre-squash policy scale $\sigma$ at the end of training. Both the fraction and the final scale increase with $M$ in every environment, though the differences in the fraction are small relative to the spread across seeds.

| | $\sigma$-increasing updates (%) | | | Policy scale $\sigma$ | | |
|---|---|---|---|---|---|---|
| | $M{=}1$ | $M{=}2$ | $M{=}4$ | $M{=}1$ | $M{=}2$ | $M{=}4$ |
| HalfCheetah | $44.5 \pm 4.3$ | $48.8 \pm 3.4$ | $50.9 \pm 3.8$ | $5.1{\times}10^{-4}$ | 0.435 | 0.769 |
| Ant | $47.7 \pm 5.8$ | $56.5 \pm 4.9$ | $57.6 \pm 5.9$ | $4.3{\times}10^{-4}$ | 0.125 | 0.211 |
| Hopper | $46.1 \pm 5.8$ | $50.1 \pm 3.9$ | $53.2 \pm 3.5$ | $6.6{\times}10^{-5}$ | 0.495 | 0.766 |

## C.5 Does ReMAC work with an SGD actor?

Since Adam's normalization interacts with the damping effect, it is natural to ask whether ReMAC depends on Adam. We reran ReMAC with a plain-SGD actor optimizer, leaving the critic optimizer, environment, learning rate, $M$, $B$, seeds, and step budget unchanged, so that the actor optimizer is the only difference; Tab. 8 reports the final returns. ReMAC does train with an SGD actor: given the substantial across-seed variability, we observe no clear separation between the two optimizers on Ant and HalfCheetah, whereas SGD performs worse on Hopper. We read this only as minimal evidence that ReMAC can work with an SGD actor, and not as a controlled comparison of the two optimizers.

Table 8: ReMAC with an Adam actor and with a plain-SGD actor; everything else, including the learning rate, is held fixed. Final return, mean $\pm$ s.d. over $n = 10$ seeds. We observe no clear separation between the two optimizers on Ant and HalfCheetah; SGD performs worse on Hopper, most markedly at $M = 4$.

| Environment | $M$ | Adam actor | SGD actor |
|---|---|---|---|
| HalfCheetah | 1 | $4562 \pm 3546$ | $4999 \pm 2742$ |
| | 2 | $4445 \pm 2973$ | $7414 \pm 4615$ |
| | 4 | $4797 \pm 3087$ | $5007 \pm 2720$ |
| Ant | 1 | $4731 \pm 586$ | $4956 \pm 667$ |
| | 2 | $5160 \pm 492$ | $5300 \pm 371$ |
| | 4 | $5382 \pm 389$ | $5351 \pm 648$ |
| Hopper | 1 | $3207 \pm 188$ | $2429 \pm 1216$ |
| | 2 | $3050 \pm 747$ | $2691 \pm 1239$ |
| | 4 | $3241 \pm 166$ | $1542 \pm 1250$ |

## C.6 Entropy of the visited-state distribution

The claims of this paper concern policy stochasticity, which policy entropy measures directly, and not exploration of the state space; we also probed the stronger notion as a check.

**Measurement.** Following prior work on maximum state-entropy exploration (Mutti et al., 2021; Seo et al., 2021), we estimate the differential entropy of the visited observation distribution using the $k$-nearest-neighbor estimator of Mutti et al. (2021) with $k = 4$. At every evaluation we roll out the current policy in 64 fresh environments, mask post-termination steps, and keep the same number of visited observations per run, pooling them into $N = 50500$ states per seed. To compare methods in a shared coordinate system, we standardize each environment's observations with a single reference mean and standard deviation, computed from a balanced pool that draws the same number of states from every method and seed. We report the results in Tab. 9.

**Result and open question.** We find no consistent improvement over SAC in the estimated entropy of the visited-state distribution: on Ant, the methods are indistinguishable within the across-seed spread, while on HalfCheetah and Hopper, SAC attains the highest mean. However, although the trend is not monotone in $M$, larger retry budgets often yield higher mean estimates than the no-retry baseline $M = 1$: this holds for $M = 2$ and $M = 4$ on HalfCheetah and for $M = 4$ on Hopper. These results provide limited additional evidence that retrying can broaden state-space visitation, while not establishing a consistent improvement over SAC. Whether ReMAC reliably improves state-space coverage therefore remains open. Settling this question requires sparse-reward or hard-exploration domains, where temporally coherent or uncertainty-directed exploration may matter more than policy stochasticity; App. C.7 reports one such evaluation.

## C.7 Sparse-reward Reacher

We also evaluate ReMAC on a sparse-reward variant of Reacher. Following the goal-conditioned convention of JaxGCRL (Bortkiewicz et al., 2025), which replaces the shaped distance reward with a binary goal indicator, we set $r_t = \mathbb{1}\left[d_t < \delta\right]$ with $\delta = 0.01$, where $d_t$ is the distance between the fingertip and the target. The

Table 9: Entropy of the visited-state distribution, estimated by the $k$-nearest-neighbor estimator of Mutti et al. (2021) with $k = 4$, mean $\pm$ s.d. over $n = 5$ seeds. Values are comparable only within an environment. On Ant the methods are indistinguishable within the across-seed spread; on HalfCheetah and Hopper SAC attains the highest mean, with no monotone trend in $M$.

| Environment | SAC | ReMAC ($M$=1) | ReMAC ($M$=2) | ReMAC ($M$=4) |
|---|---|---|---|---|
| Ant | $21.20 \pm 1.56$ | $21.15 \pm 1.39$ | $21.92 \pm 1.78$ | $20.74 \pm 2.15$ |
| HalfCheetah | $11.05 \pm 1.95$ | $-4.15 \pm 5.98$ | $0.52 \pm 5.91$ | $7.69 \pm 1.72$ |
| Hopper | $2.21 \pm 0.75$ | $-1.98 \pm 1.76$ | $-2.05 \pm 5.78$ | $-0.68 \pm 6.62$ |

return is therefore the number of steps of the 50-step episode spent within $\delta$ of the goal; a uniformly random policy attained an average return of 0.15 over 2048 episodes in our measurement. All other settings are those of the dense Reacher.

Tab. 10 reports the results. Every $M > 1$ attains a higher average return than $M = 1$, which is also the most variable configuration across seeds, and attains mean returns comparable to SAC.

Table 10: Return on the sparse-reward Reacher ($\delta = 0.01$; the maximum is 50, and a uniformly random policy attained 0.15 over 2048 episodes). Final values, mean $\pm$ s.d. over $n = 10$ seeds.

| | Return |
|---|---|
| SAC | $29.50 \pm 5.37$ |
| ReMAC ($M = 1$) | $24.02 \pm 8.86$ |
| ReMAC ($M = 2$) | $30.73 \pm 5.04$ |
| ReMAC ($M = 4$) | $31.21 \pm 4.77$ |
| ReMAC ($M = 8$) | $28.02 \pm 5.01$ |

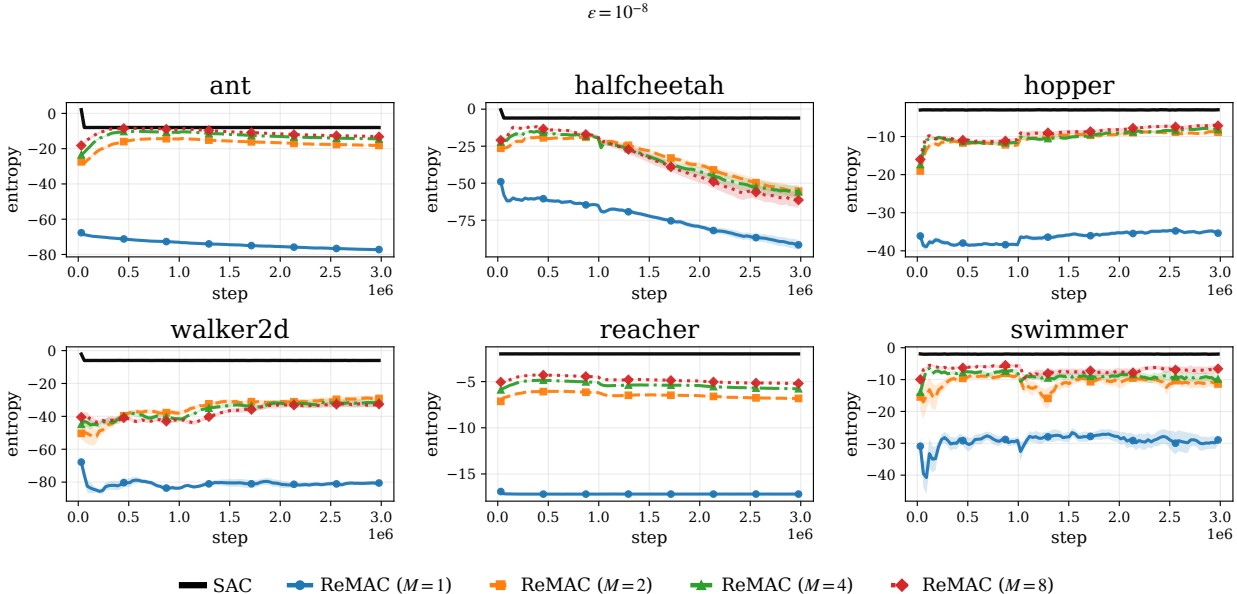

Figure 7: The average policy entropy of ReMAC with $M \in \{1, 2, 4, 8\}$ at Adam's default $\varepsilon = 10^{-8}$, for all tasks. Within a panel, $M$ carries the color that Figs. 3 and 4 assign to it, together with a distinct line style and marker; SAC is shown for reference. For $M = 1$, entropy increases with $\varepsilon$ in every task (Figs. 7–10).

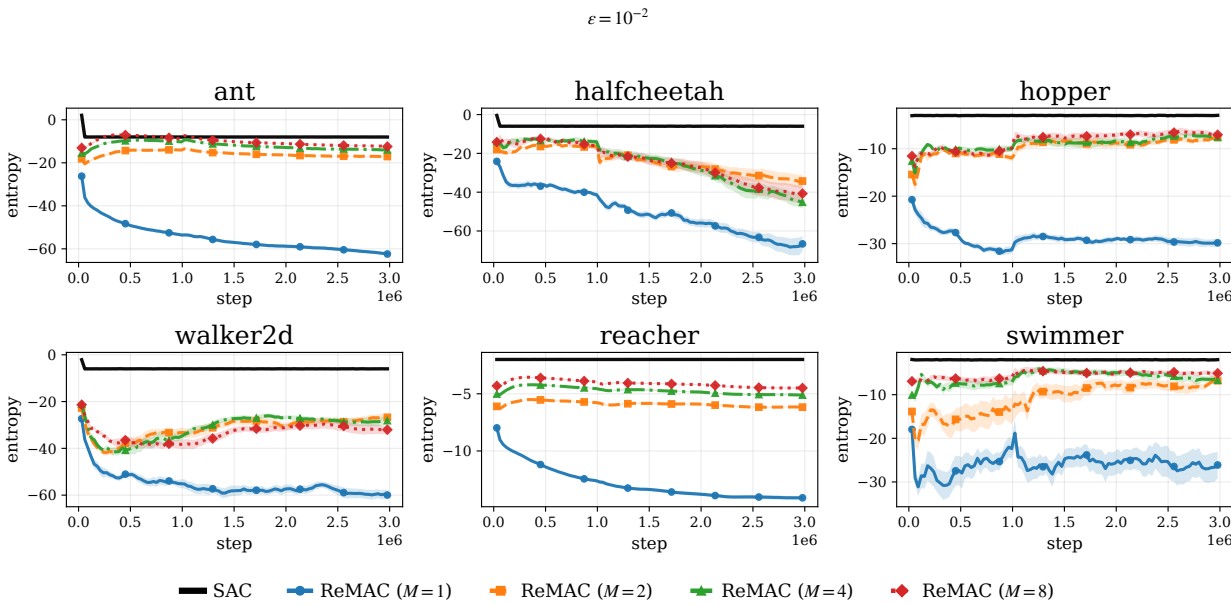

Figure 8: The average policy entropy of ReMAC with $M \in \{1, 2, 4, 8\}$ at $\varepsilon = 10^{-2}$, for all tasks. Same encoding as Fig. 7.

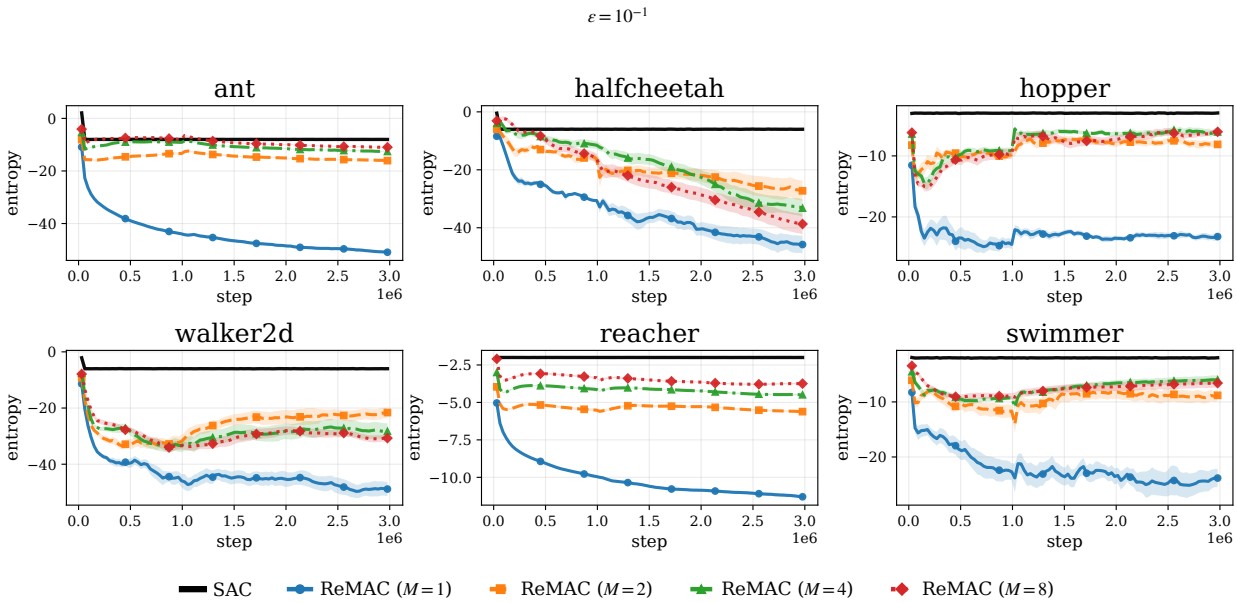

Figure 9: The average policy entropy of ReMAC with $M \in \{1, 2, 4, 8\}$ at $\varepsilon = 10^{-1}$, for all tasks. Same encoding as Fig. 7.

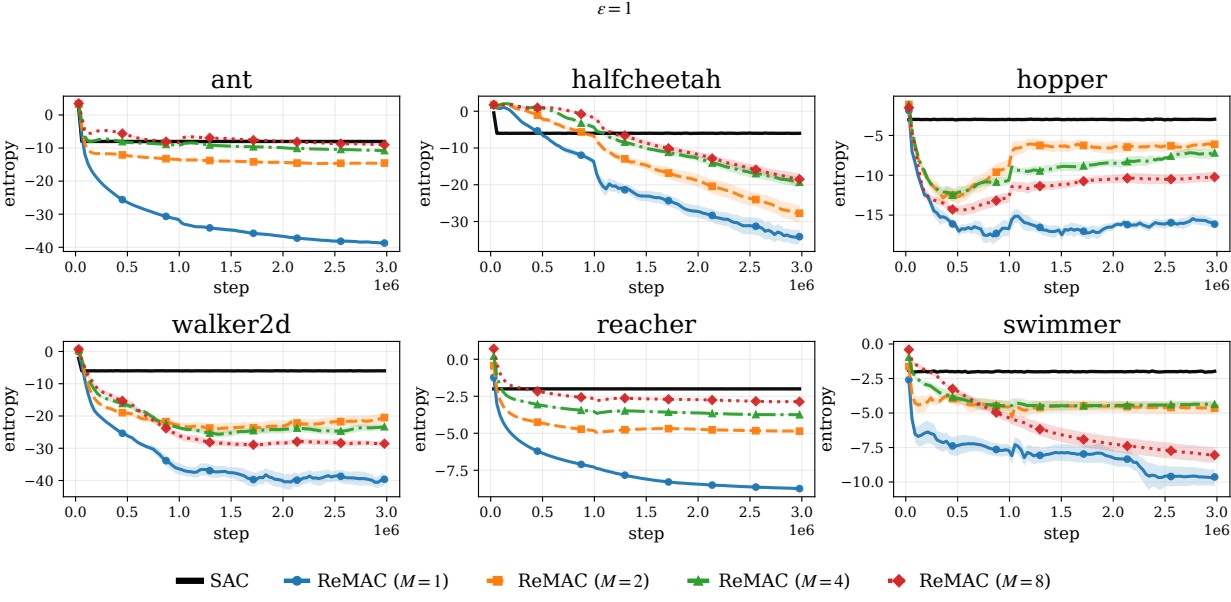

Figure 10: The average policy entropy of ReMAC with $M \in \{1, 2, 4, 8\}$ at $\varepsilon = 1$, for all tasks. Same encoding as Fig. 7.

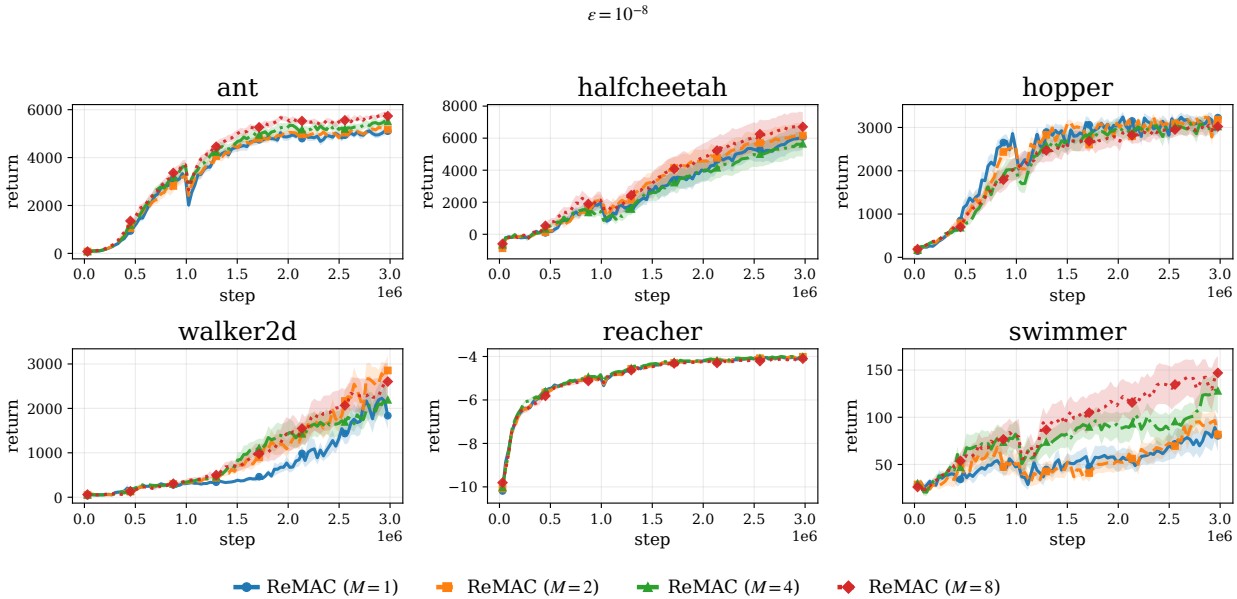

Figure 11: The average return of ReMAC with $M \in \{1, 2, 4, 8\}$ at Adam's default $\varepsilon = 10^{-8}$, for all tasks. Same encoding as Fig. 7, without the SAC reference.

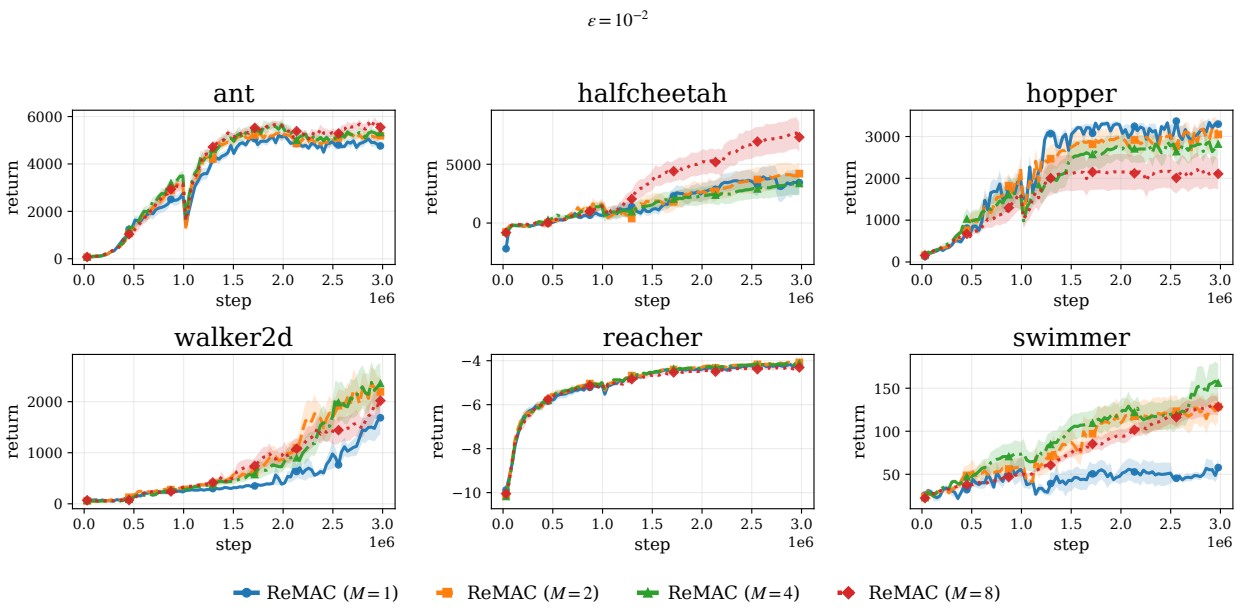

Figure 12: The average return of ReMAC with $M \in \{1, 2, 4, 8\}$ at $\varepsilon = 10^{-2}$, for all tasks. Same encoding as Fig. 11.

$\varepsilon = 10^{-1}$

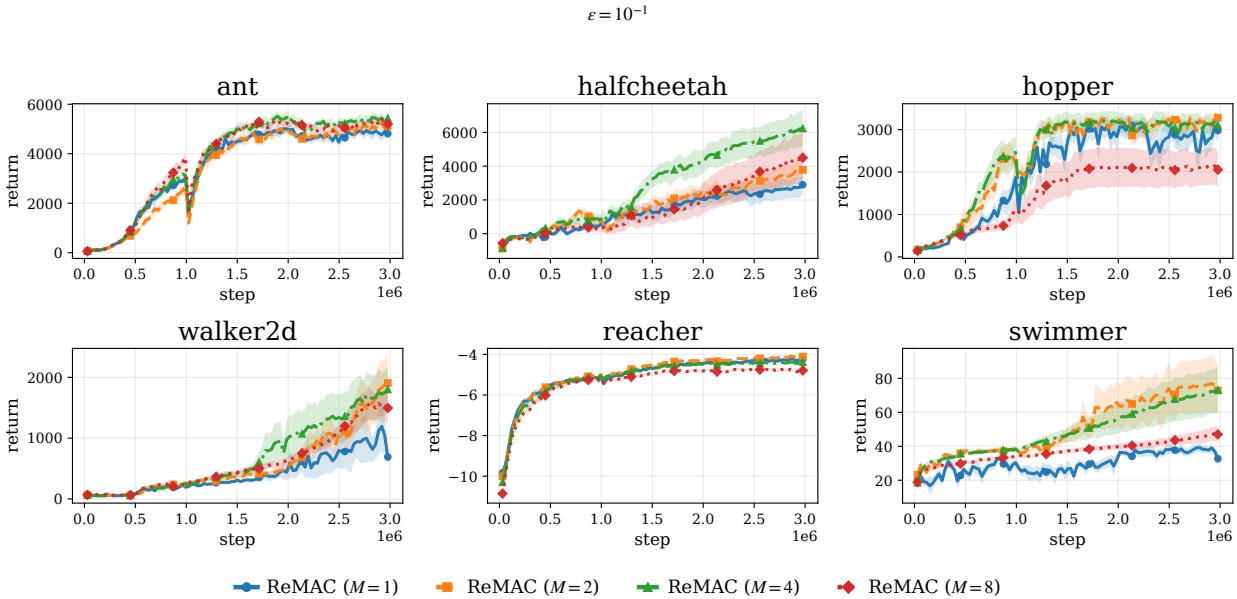

Figure 13: The average return of ReMAC with $M \in \{1, 2, 4, 8\}$ at $\varepsilon = 10^{-1}$, for all tasks. Same encoding as Fig. 11.

$\varepsilon = 1$

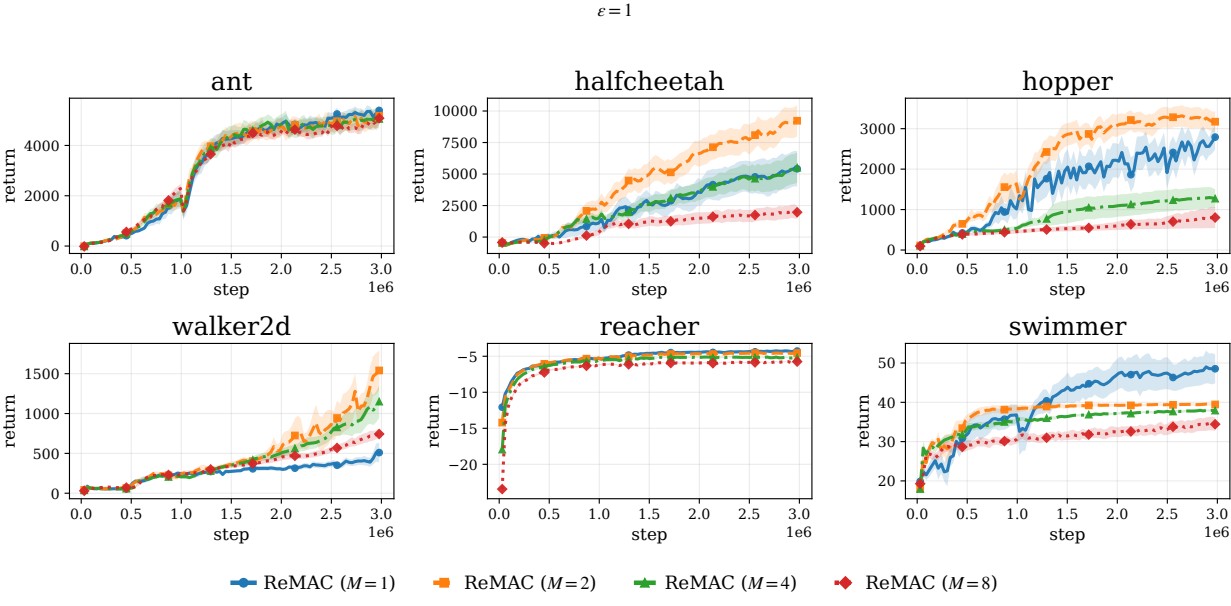

Figure 14: The average return of ReMAC with $M \in \{1, 2, 4, 8\}$ at $\varepsilon = 1$, for all tasks. Same encoding as Fig. 11.

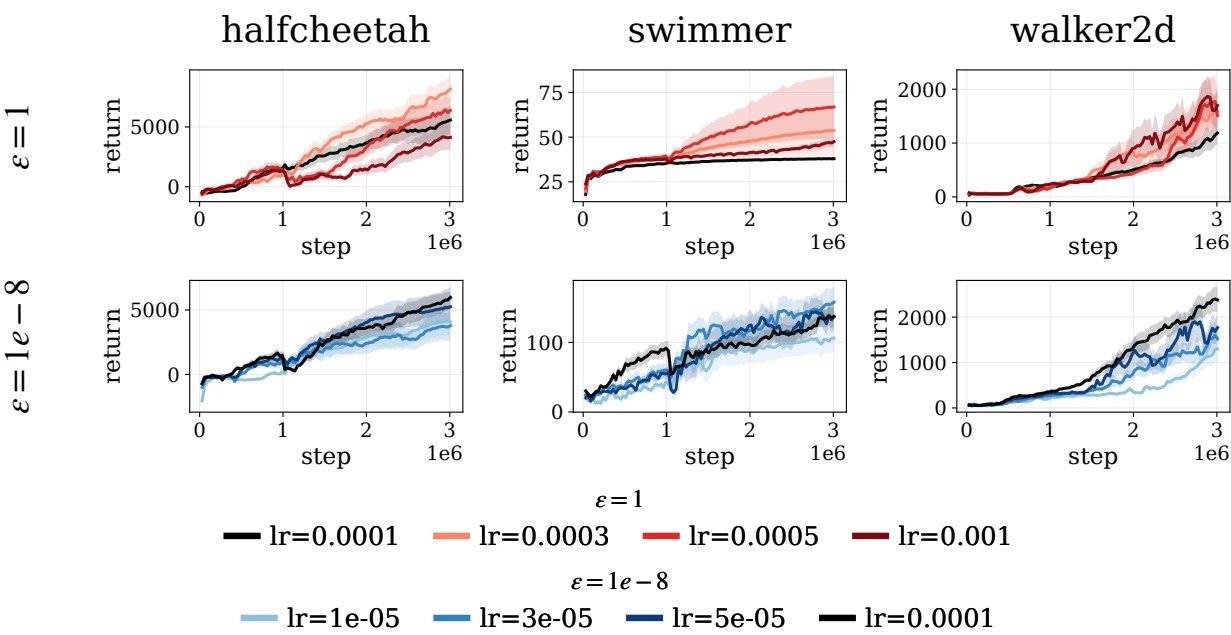

Figure 15: The average return of ReMAC with $M = 4$ under different learning rates: $\varepsilon = 1$ (top row, $\alpha \in \{10^{-4}, 3 \times 10^{-4}, 5 \times 10^{-4}, 10^{-3}\}$) and $\varepsilon = 10^{-8}$ (bottom row, $\alpha \in \{10^{-5}, 3 \times 10^{-5}, 5 \times 10^{-5}, 10^{-4}\}$), for HalfCheetah, Swimmer, and Walker2d. The shared default learning rate $\alpha = 10^{-4}$ is drawn in black in both rows.

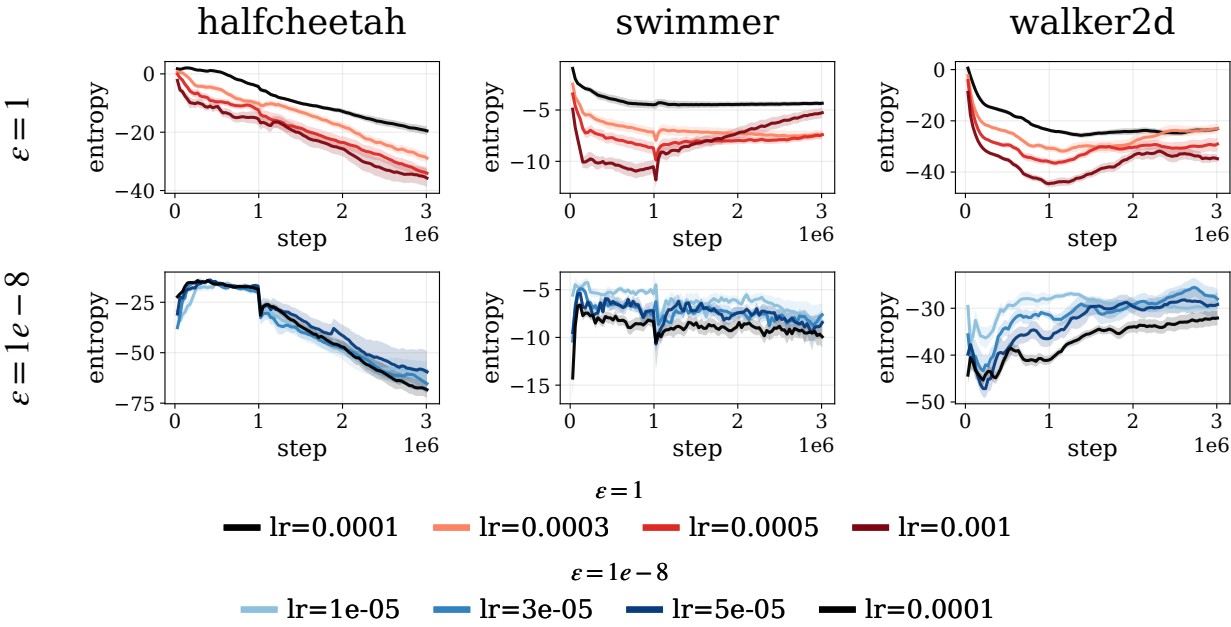

Figure 16: The average policy entropy of ReMAC with $M = 4$ under different learning rates: $\varepsilon = 1$ (top row, $\alpha \in \{10^{-4}, 3 \times 10^{-4}, 5 \times 10^{-4}, 10^{-3}\}$) and $\varepsilon = 10^{-8}$ (bottom row, $\alpha \in \{10^{-5}, 3 \times 10^{-5}, 5 \times 10^{-5}, 10^{-4}\}$), for HalfCheetah, Swimmer, and Walker2d. The shared default learning rate $\alpha = 10^{-4}$ is drawn in black in both rows.

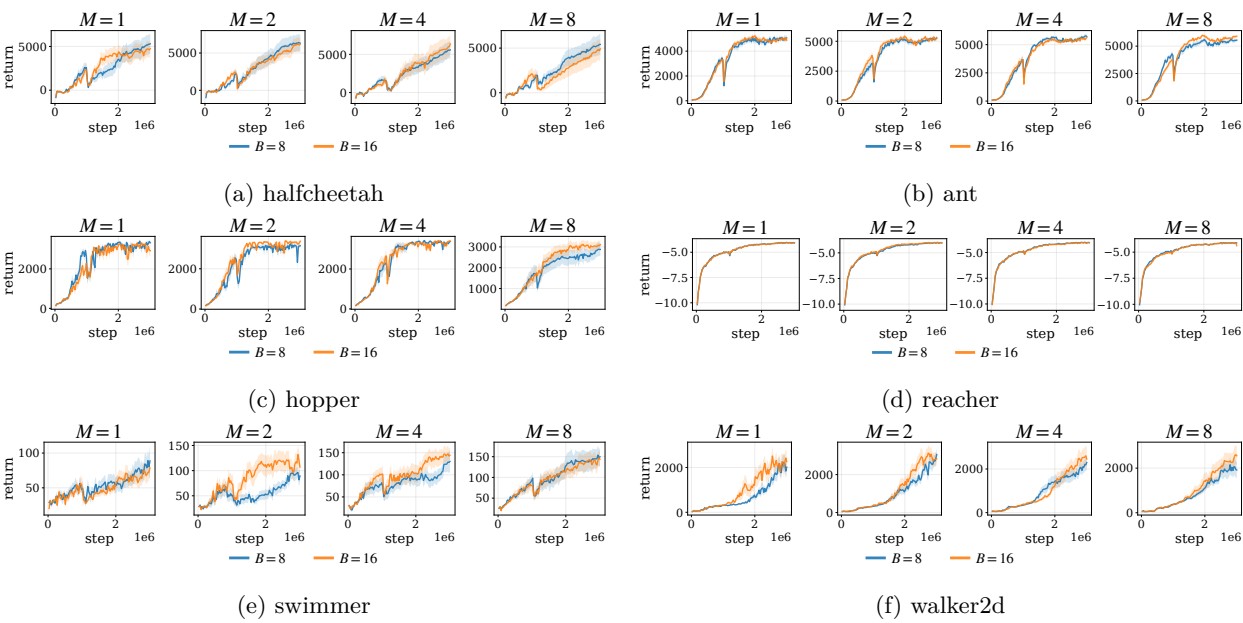

Figure 17: The average return of ReMAC with $M = 1, 2, 4, 8$ and $\varepsilon = 10^{-8}$ under different batch sizes $B \in \{8, 16\}$ for all tasks.

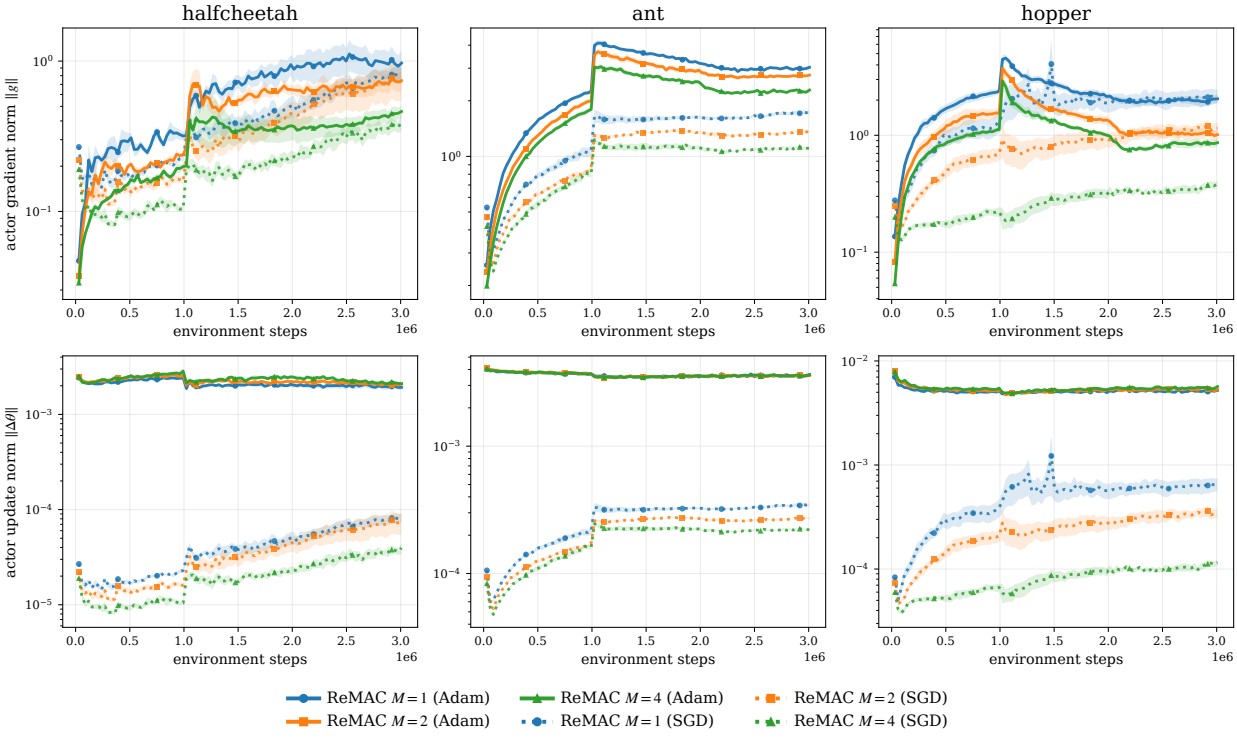

Figure 18: **Gradient damping on Brax.** Top: the actor gradient norm. Bottom: the realized update norm. Solid lines use the Adam actor, dotted lines the SGD actor; shading is the standard error over 10 seeds. The gradient norm shrinks with $M$ in every environment and for both optimizers, consistent with the damping mechanism suggested by Proposition 3, Corollary 1, and the toy analysis. The update norm follows the gradient norm under SGD but not under Adam, whose per-coordinate normalization divides out the scale of the gradient—the effect analyzed in Sec. 5.3. Runs for different $M$ are trained separately, so this comparison is not a controlled isolation of the retry weighting.

