# OpenReview forum: "Retry Policy Gradients in Continuous Action Spaces"
_TMLR — Under review for TMLR_

### Review · Reviewer_cxT2 · 2026-06-26

**Summary Of Contributions:**

The paper generalises a Retry strategy (also referred as ReMax) to Actor-Critic algorithms like SAC, which is presented as an alternative to entropy regularisation. The retry strategy defines a new policy objective that maximises a Q function computed with the best out of M actions sampled at random from the policy. In practice, this involves the special reparametrization gradient estimator for policy proposed by Walder&Karkhanis (2025). The authors have analysed the effect of the new objective on the simple toy environment, governed only by an action cost function. Theoretically, this allows authors to establish the entropy increase effect with M>1. Practically, ReMAC manages to achieve similar performances to SAC algorithm on a few environments from Brax (MuJoCo like problems).

The strengths of the paper:
- This paper is well-written with clear motivation and explanation behind the proposed algorithmic improvement.
- The toy environment clearly demonstrates the properties of the ReMax in continuous action setting. Those properties are entropy increase for M>1 and dampening of the gradients near optimum.
- The experiments on a more difficult continuous-action environment show that ReMAC is competitive with SAC, without entropy bonus.

The weaknesses
- It is not clear if theoretical results proposed in this paper can be easily generalized outside the setting studied in the paper (single-step optimization, reward independent from state).
- The experiments on Brax tasks remain quite limited, e.g. there is no humanoid task, while it is also available in Brax and quite standard for testing continuous-action algorithms.

**Audience:**

Yes

**Audience Explanation:**

Yes, the paper clearly falls in the domain of new algorithms for reinforcement learning.

**Claims And Evidence:**

Yes

**Claims Explanation:**

Yes, the claims made in submission are supported by both theoretical analysis and practical experiments on several environments with continuous action setting. Nonetheless, the theoretical analysis remains quite limited, as only one simple setting is studied, where value function is defined with a single reward depending only on the action. While this toy example presents interesting insights, it remains mostly motivational. Therefore, the statements in abstract "We study the resulting learning dynamics…" can be perceived in a more general sense than it actually is. Despite that, this theoretical analysis is accompanied with experiments on a few complex continuous-action environments that show empirically a positive effect of retry strategy on performance and entropy.

**Requested Changes:**

- Add experiments on Humanoid, a more challenging benchmark but also available in Brax.
- In your tests, how is entropy regularization implemented in SAC? Do you use a fixed entropy bonus? annealing? or automatic tuning? On this note, SAC’s performance on HalfCheetah looks lower than usual.
- Add a discussion, on how to choose M. What are the tradeoffs as It seems none of the choices of M is behaving consistently well across all environments?
- For extra figures 6-13, please choose better colormaps, it is hard to distinguish one curve from another. Instead of having one figure per M, consider having one figure per epsilon with varying M to see more clearly if larger M implies bigger entropy bonus.
- Does ReMAC works with SGD as an optimizer on Brax environments? Is it comparable to Adam under some choice of epsilon?
- Is dampening effect verified in practice on Brax environments?
- (optional) verifying the entropy increase effect through gradients with respect to sigma on Brax environments?

---

> ### Author Response · Authors · 2026-07-15
>
> Thank you very much for the review. Main revisions: we added HumanoidStandup, an SGD comparison, Brax gradient measurements, a scale-gradient proxy, and the requested figure reorganization. Changes are marked in **magenta** in the revised paper.
> We address the requested changes below.
>
> ### 1. Humanoid experiment
> Because rejax does not provide a tuned configuration for Humanoid, we evaluated **HumanoidStandup**, the only humanoid-class task for which rejax provides a tuned configuration and our highest-dimensional environment ($d=17$; App. C.3, Tab. 4). **Mean return increases from $25209\pm1138$ at $M=1$ to $27575\pm497$ at $M=8$, with $M\ge2$ exceeding SAC's $25427\pm1081$.**
>
> **On the different $\epsilon$ values (from the discussion).** We ran HumanoidStandup at $\epsilon\in\{10^{-8},10^{-1},1\}$ (new App. C.3, Tab. 5,). At the default $\epsilon=10^{-8}$, larger $M$ does *not* reliably raise entropy here — $M=4,8$ are near or below $M=1$ — because Adam's normalization counteracts the ReMax entropy-increase effect. As $\epsilon$ grows this normalization weakens and the effect becomes visible: at $\epsilon=1$ every $M>1$ exceeds $M=1$ (entropy $-128,-70,-79,-87$ for $M=1,2,4,8$), and overall entropy rises with $\epsilon$ at every $M$.
>
> Overall, this additional experiment directly addresses the concern about evaluation on a challenging humanoid benchmark: on HumanoidStandup, ReMAC with $M>1$ consistently achieves higher mean returns than $M=1$ across all tested values of $\epsilon$.
>
> ### 2. SAC entropy regularization and HalfCheetah
>
> SAC uses automatic temperature tuning: $\alpha$ is learned from an initial value of 1 toward target entropy $-\dim(\mathcal A)$, rather than fixed or annealed. Sec. 7 now states this.
> On the HalfCheetah return of SAC, we believe it is in the normal range for this benchmark. **For reference, please see Figure 4 of Brax paper (arXiv:2106.13281)**.
>
> ### 3. Choosing $M$
> Performance is not monotone in $M$, and no single value is best across environments, while $M>1$ generally matches or improves on $M=1$, and $M=8$ has higher final entropy than $M=1$. We therefore treat $M$ as a stochasticity-control hyperparameter and recommend $M\in\{2,4\}$ given the cost constraint $B\ge M$.
>
> ### 4. Figure layout
> Figs. 7--14 now fix $\epsilon$ and compare $M\in\{1,2,4,8\}$ within each panel using distinct colors, line styles, and markers. Figs. 7--10 report entropy and Figs. 11--14 return.
>
> ### 5. SGD comparison and the role of $\epsilon$
> We reran ReMAC with a plain-SGD actor while holding all other settings fixed (App. C.5, Tab. 8). Adam and SGD achieve comparable mean returns on HalfCheetah and Ant, while SGD is lower on Hopper, especially at $M=4$. **Thus ReMAC can train with SGD**, although performance depends on the optimizer--learning-rate pairing; the Hopper learning rate was tuned for Adam and reused for SGD.
>
> For Adam, if $\sqrt{\hat v_t}\ll\epsilon$, the update approaches momentum SGD with effective learning rate $\alpha/\epsilon$, not plain SGD. In the toy problem, $\epsilon=1$ empirically tracks SGD. On Brax, larger $\epsilon$ tend to raise entropy and sometimes lowers return, while increasing $\alpha$ at $\epsilon=1$ recovers much of the lost return.
>
> ### 6. Gradient damping on Brax
> We logged **actor-gradient and realized-update norms** on HalfCheetah, Ant, and Hopper (App. C.4, Fig. 18, Tab. 6). At $M=4$, the gradient norm is 17--75% of its $M=1$ value. Under SGD, the update ratio follows the gradient ratio; under Adam, update ratios remain near one, **consistent with adaptive normalization removing much of the gradient-scale effect**.
>
> ### 7. (Optional) Verify the entropy-increase effect through $\sigma$-gradients on Brax
> We added this analysis in **App. C.4 and Tab. 7**. We log the actor-loss gradient w.r.t. $b_\sigma$, the bias of the log-scale head; a **negative** value pushes the pre-squash scale up. We report the *fraction* of such updates. That fraction rises with $M$ in all three environments, as does the final scale (2–4 orders of magnitude). Together, these results provide empirical evidence that the entropy-increase mechanism identified in Proposition 1 has a measurable analogue in practical Brax training.

---

> ### Author Response · Authors · 2026-07-15
>
> ### Generality of the theory
>
> Sec. 5.2 now states that the analysis concerns a fixed-state, fixed-critic objective with scalar scale and direct parameters, rather than coupled actor--critic dynamics. Two points on how it extends. First, the gradient with respect to the actual policy parameters $\theta$ follows from the $(\mu,\sigma)$-gradients by the chain rule: adding the Jacobian $\partial(\mu_\theta,\sigma_\theta)/\partial\theta$ maps our results to $\theta$, and the entropy-increase conclusion is preserved. Second, the full actor objective is an expectation over states, so it is the state-average of the fixed-state problem we analyze, and the mechanism remains valid on average across states. The idealization is in the shape of the cost; the Brax measurements provide evidence consistent with the isolated mechanism. We added a short sentence to this effect in Sec. 5.2.
>
> ### Clarification on the gradient estimator
> One attribution point: the reparameterization (pathwise) ReMax estimator we use is proposed by us, not by Walder \& Karkhanis (2025), who introduced a likelihood-ratio-type estimator for the retry objective.
> We thank the reviewer again for the constructive feedback, which helped us substantially improve the paper.

---

### Review · Reviewer_22h8 · 2026-07-02

**Summary Of Contributions:**

The paper investigates retry-based objectives (ReMax) for continuous-control reinforcement learning. It argues that ReMax promotes stochastic exploration by simultaneously encouraging policy updates toward higher-entropy regions while reducing gradient magnitudes, thereby slowing policy convergence. The paper further analyzes how adaptive optimizers such as Adam alleviate the resulting gradient attenuation through gradient normalization. Building on this analysis, the authors instantiate the ReMax objective within the Soft Actor-Critic (SAC) framework, resulting in the proposed ReMAC algorithm. The approach is evaluated on a suite of continuous-control benchmarks and compared against SAC and PPO.

**Additional Comments:**

n/a

**Audience:**

Yes

**Audience Explanation:**

**Yes.**

The paper addresses an interesting research question, and the proposed idea has the potential to make a meaningful contribution. However, the current presentation makes the paper difficult to follow, and several key claims are not sufficiently supported by either the theoretical analysis or the empirical evidence. In particular, the assumptions underlying the analysis appear overly simplified relative to the actual reinforcement learning setting, making it difficult to assess whether the conclusions generalize to the proposed algorithm. Overall, while the work is promising, the paper would benefit from substantially clearer exposition, stronger theoretical justification, and more convincing empirical validation before its claims can be fully assessed.

**Claims And Evidence:**

No

**Claims Explanation:**

Several of the paper's central claims are not sufficiently supported, partly because the theoretical analysis is conducted under highly simplified settings that do not clearly justify the conclusions drawn for the proposed algorithm.

1. **"The ReMax gradient encourages stochastic exploration in both direction and magnitude when ($M>1$)."**
   This claim appears overstated. Specifically:

   * Figure 1 shows that the gradient with respect to ($\sigma$) points toward increasing ($\sigma$) only when ($\sigma$) is small, while it remains nearly unchanged when ($\sigma$) is already large. The latter behavior does not appear to encourage additional exploration.
   * The changes in gradient magnitude shown in Figure 1 are relatively subtle, making it difficult to conclude that they meaningfully contribute to increased exploration.

   More importantly, the analysis in Section 5.2 is derived under an overly simplified setting where the reward (or cost) depends only on the action and the decision problem consists of a single step. This setting differs substantially from the policy-gradient objective in Equation (2), making it unclear whether the conclusions regarding the "ReMax gradient" generalize to the actual reinforcement learning setting.

2. **"The gradient norm diminishes near the optimum."**
   This claim is not clearly established by the theoretical analysis. In Proposition 3 (Gradient Damping), the upper bounds on the gradients with respect to both ($\mu$) and ($\sigma$) depend only on the action norm and are independent of the policy parameters themselves. Consequently, the proposition does not demonstrate that the gradient norm decreases as the policy approaches an optimum. As presented, Proposition 3 does not appear to provide theoretical evidence for the claimed gradient-damping phenomenon near convergence.

3. **"Adam's adaptive normalization can mitigate this damping."**
   The evidence supporting this claim is not fully convincing. The analysis modifies Adam by changing the numerical stabilization parameter ($\epsilon$), but this does not isolate the effect of adaptive normalization. Gradient normalization remains present regardless of the particular choice of ($\epsilon$); changing ($\epsilon$) merely alters the extent of the normalization rather than removing it. A more appropriate comparison would be either:

   * comparing Adam with a version that removes the denominator entirely (i.e., using only the exponential moving average of gradients without adaptive normalization), or
   * directly analyzing the normalized ReMax gradients under Adam (or AdaGrad-like normalization).

   Furthermore, the analysis does not discuss the influence of the bias-correction terms associated with ($\beta_1$) and ($\beta_2$), which may further complicate the behavior of the effective gradient magnitude.

4. **"ReMax Actor-Critic (ReMAC), an off-policy actor--critic algorithm that optimizes the ReMax objective using a pathwise derivative estimator."**
   The implementation appears closer to a modified version of SAC than to a fundamentally new optimization procedure. In particular, ReMAC is implemented by removing the entropy regularization term from SAC while retaining the remaining training pipeline. It is therefore somewhat confusing to describe the method as optimizing the ReMax objective using a pathwise derivative estimator, since this is already the standard gradient estimator employed in SAC. This interpretation is also consistent with the empirical results in Figure 3, where ReMAC and its variants exhibit nearly identical learning dynamics to SAC across several environments (e.g., Ant, HalfCheetah, and Hopper), suggesting that the optimization procedure itself remains largely unchanged.

**Requested Changes:**

### Required Adjustments

1. **Improve the consistency and precision of the notation.**

   The paper frequently mixes the notation for scalars, vectors, and functions, making the theoretical analysis difficult to follow and, at times, less convincing.

   * Throughout the paper, ($\mu$) and ($\sigma$) are used to denote both scalars and vectors. For example, they are treated as scalars in Section 5.1 but as vectors in Section 5.2. This is particularly problematic for ($\sigma$), since it is unclear whether a single scalar variance is assumed for all action dimensions. In practical implementations such as SAC, however, ($\sigma$) is parameterized as a state-dependent vector rather than a scalar.

   * The notation in the main theoretical results is also imprecise. In Proposition 1, ($\partial_{\sigma}J^M$) should represent the gradient with respect to a vector parameter, rather than a scalar quantity, making the inequality
     $$\partial_{\sigma}J^M > 0$$
     mathematically ambiguous. The same issue appears in Proposition 2, where ($\partial_{\sigma}J^1$) should likewise be a vector. It is also unclear what the notation ($d\sigma$) refers to in this proposition. In Proposition 3, the norm used in the gradient bound is not explicitly defined, while an absolute value is used for ($\partial_{\sigma}J^1$), resulting in inconsistent treatment of vector quantities.

   * The notation $A$ is overloaded. In the likelihood-ratio policy gradient formulation, $A$ denotes the advantage function, whereas in the setup and assumptions the same symbol is reused to denote sampled actions. This reuse is confusing, particularly since the paper already uses $a$ to represent actions elsewhere.

   * The notation for the reward (or cost) function also changes throughout the paper. It is initially defined as $\mathcal{R}(s,a)$, but the theoretical analysis later switches to $r(a)$ (or $c(a)$), effectively removing the state dependence without explicitly explaining this simplification. This change makes it more difficult to understand the relationship between the theoretical analysis and the original reinforcement learning objective.

### Important Questions

1. **What exactly is meant by the "policy scale"?**
   The paper repeatedly refers to situations where "the policy scale is small," but this quantity is never formally defined. It is therefore unclear whether the policy scale refers to the standard deviation $\sigma$, the action variance, the policy entropy, or another quantity.

2. **Why does $\epsilon = 1.0$ correspond to SGD?**
   The paper states that setting the numerical stabilization parameter to $\epsilon=1.0$ aligns Adam with SGD. This statement is not immediately obvious and would benefit from additional explanation or theoretical justification.

3. **Why does the policy entropy continue increasing during training in some environments?**
   In Figure 4, the policy entropy for Hopper and Walker2d appears to increase throughout training, eventually exceeding its initial value. This behavior seems counterintuitive and would be useful to explain, particularly given the paper's discussion of exploration and gradient damping.

---

> ### Author Response · Authors · 2026-07-15
>
> Thank you very much for the review.
> We have revised the paper to address the reviewer's concern. All changes are marked in **blue**．
>
> ### W1. Direction and magnitude of the ReMax gradient
>
> > Figure 1 shows that the gradient with respect to $\sigma$ points toward increasing $\sigma$ when $\sigma$ is small...
>
>  When $\sigma$ is already large, the policy is sufficiently stochastic, so the near-flat $\sigma$-gradient is desirable rather than a shortcoming.
>
> >The changes in gradient magnitude shown in Figure 1 are relatively subtle...
>
> The magnitude differences are not subtle. In the toy vector field (Fig. 1) the gradient norm varies by up to two orders of magnitude ($>100\times$) across $M$.
>
> > More importantly, the analysis in Section 5.2 is derived under an overly simplified setting ...
>
> Our analysis is intended to isolate the mechanism behind the ReMax actor update, rather than to model the full actor--critic dynamics.
> That said, for a fixed state and critic, setting $c(a)=-Q(s,a)$ reduces the actor objective to the cost studied in Sec. 5.
> The full ReMAC objective additionally averages over states and involves a learned critic and neural-network policy.
> We now state these limitations explicitly in Sec. 5.2.
>
> ### W2. Gradient norm near the optimum
>
> The apparent discrepancy comes from interpreting the policy parameters as the neural-network parameters $\theta$.
> In Proposition 3, $A_m=\mu+\sigma\xi_m$, so the bounds goes to zero as $(\mu,\sigma)\to(0,0)$.
> **For a neural policy, the gradient with respect to $\theta$ is obtained through the Jacobian** of $(\mu_\theta,\sigma_\theta)$; therefore, if its operator norm remains bounded, the vanishing $(\mu,\sigma)$-gradients imply a vanishing full policy-parameter gradient. We added this clarification to Sec. 5.2.
>
> ### W3 Adam's adaptive normalization
> To isolate the effect of adaptive normalization, we compared Adam, SGD, and an EMA-only update in App. B.2, Fig. 6, and Tab. 1. The EMA-only update is
> $\theta_{t+1}=\theta_t-\alpha\hat m_t,$
>
> which retains the first-moment EMA and its bias correction while removing the second-moment denominator entirely. Thus, $\beta_2$ and its bias correction play no role in this variant. At $M=8$, the slowdown factors are 7584 for EMA-only, 7614 for SGD, 7426 for Adam with $\epsilon=1$, 418 for $\epsilon=10^{-1}$, and 1.9 for $\epsilon=10^{-8}$. Removing the denominator therefore reproduces SGD-like damping, **whereas default Adam removes most of it**.
>
> We also compared raw actor gradients and realized updates on **Brax**. Under Adam, the $M=4$/$M=1$ gradient ratios are 0.45/0.75/0.43, while the update ratios are 1.10/1.00/1.07; under SGD, both are 0.46/0.65/0.17. **This is consistent with the adaptive denominator removing the gradient-scale effect**.
>
> ### W4 ReMAC versus SAC
>
> The pathwise actor gradient is not specific to SAC; it was already used in deterministic policy-gradient methods, while SAC combines this gradient principle with entropy regularization. We therefore view ReMAC not as SAC without entropy regularization, but as the pathwise-gradient instantiation of the ReMax objective. ReMAC builds on SAC for critic learning and implementation convenience, but its actor objective is different.
> Therefore, our references to SAC are a practical, implementation-level framing rather than an algorithmic one.
> The similar curves in Fig. 3 indicate comparable performance, not identical actor updates.

---

> ### Author Response · Authors · 2026-07-15
>
> ### Q1. What is the policy scale?
>
> It is the scalar standard deviation $\sigma$ of the isotropic Gaussian used in the theory. ReMAC instead uses a state-dependent diagonal scale $\sigma_\theta(s)$. Sec. 5 now defines both explicitly, and we added short reminders of the $\sigma$ / $\sigma_\theta(s)$ distinction at a few more places so it is not missed.
>
> ### Q2. Why does $\epsilon=1$ correspond to SGD?
>
> Strictly, it does not. If $\sqrt{\hat v_t}\ll\epsilon$, then
>
> $\frac{\hat m_t}{\sqrt{\hat v_t}+\epsilon}
> \approx
> \frac{\hat m_t}{\epsilon},$
>
> so large-$\epsilon$ Adam approaches momentum SGD with effective learning rate $\alpha/\epsilon$. In the toy problem, $\epsilon=1$ empirically tracks the plain-SGD baseline, but 1 has no special theoretical status.
>
> ### Q3. Why does entropy keep increasing in Hopper and Walker2d?
>
> Our theory does not explain persistent entropy growth. Proposition 1 is local in the policy scale. A plausible hypothesis is that the moving critic, replay distribution, and state distribution keep the actor objective non-stationary, but we have not isolated this experimentally and do not present it as a theoretical prediction.
>
> ### Notation
>
> Sec. 5 now defines $\mu\in\mathbb{R}^d$ as a vector and $\sigma>0$ as a scalar, so $\nabla_\mu$ is vector-valued and $\partial_\sigma$ is scalar-valued. ReMAC's diagonal $\sigma_\theta(s)\in\mathbb{R}^d$ is distinguished from the theoretical scalar. Proposition 2 now uses $-(\lambda d)\sigma$; advantages are denoted $\widehat{\mathrm{Adv}}$; $c(a)=-Q(s,a)$ explains the fixed-state reduction; Gaussian noise is denoted by $\xi$, reserving $\epsilon$ for Adam; and $Z_m$ and $\beta_M$ are defined at first use.
>
> We thank the reviewer again for the constructive feedback, which helped us substantially improve the paper.

---

### Review · Reviewer_Lak5 · 2026-07-02

**Summary Of Contributions:**

This paper extends retry-based policy-gradient objectives from discrete action spaces to continuous action spaces, and studies why such objectives can induce exploration without adding explicit entropy bonuses.  Theoretically, the paper provides some theoretical explanations under stateless MDP and deterministic reward, with some additional assumptions. Empirically, it shows that the proposed method ReMAC  achieves performance comparable to entropy regularized method SAC.

Strength
- The paper is generally well written is easy to follow, except some minor typos. The vector field plot in the introduction is an good example to highlight the key insights of this paper. The related work is also well written.
-  One key finding is how retry objective reshapes gradient landscape compare with entropy regularization. This is an interesting conceptual contribution.
- The algorithm implementation is simple, which only modifies SAC by removing entropy terms and replacing the actor loss with a ReMax loss.
- The theory restricted to the toy case is very clean.

Weakness
- The theory is quite stylized as it rely on smoothness, strong convexity, deterministic rewards, isotropic Gaussian policies, and local analysis, especially the strong convexity. Among these assumpstions, I believe the strong convexity is the major one that is far from typical deep RL settings.
- The experiments mostly measure policy entropy, but entropy alone is a weak proxy for meaningful exploration. The paper does not convincingly demonstrate improved state-space coverage and sparse-reward performance.
- The computational cost of REMAC seems nontrivial. I noticed that Table 2 indicated B = 16 can cause double training time of REMAC comapared to MAC.

**Additional Comments:**

additional questions/comments
- does the performance increase in $M$? if not, what is the best choice of $M$ given the computation-performance tradeoff?
- Prop 3: I think the two expectation terms in the right hand side can be further bounded in terms of $\mu, \sigma, M$. It would be good to establish the dependence of the upper bound on $M$.
- equation 4: is $\epsilon_{1:B}$ scalar or vector?


some typos
- Page 6, $Z_m$ undefined

**Audience:**

Yes

**Audience Explanation:**

Exploration is one of the most research problem in RL community. The paper focuses on the stochastic exploration, which I believe aligns well with general RL research community. It provides useful insights into how retry-based objectives behave in continuous action spaces, including gradient landscape and entropy. So it may interest researchers working on exploration, entropy regularization, and continuous-control actor–critic methods.

**Claims And Evidence:**

No

**Claims Explanation:**

The claims are partially supported. The theoretical and toy-gradient evidence indeed demonstrates that ReMax can reshape the gradient landscape in continuous action spaces.  ReMAC is also empirically shown to achieve performance comparable to SAC. However, the evidence for exploration in the stronger RL sense is limited. The experiments mostly report entropy and returns on dense-reward tasks, without measuring state coverage and/or sparse-reward performance.

**Requested Changes:**

Adding experiments that measures state coverage and/or sparse-reward performance of REMAC would greatly strengthen the paper.

---

> ### Author Response · Authors · 2026-07-15
>
> Thank you very much for the review. We added an explicit bound for Proposition 3 (**new Corollary 1**) and a state-coverage evaluation (**new App. C.6**) and simple sparse task evaluation (**new App.C.7**). Changes are marked in **green** in the revised paper.
>
> ### W1. Stylized theory and strong convexity
> The analysis is stylized, but smoothness and strong convexity are standard assumptions in first-order optimization theory [1] and provide a natural first setting for isolating the ReMax mechanism.
>
> **Our aim is to build intuition rather than model the full RL objective**.
> The full actor objective is an expectation over states, and at each state $s$, with the critic fixed, it reduces to the action-cost problem analyzed with $c(a)=-Q(s,a)$.
> The central restrictive assumption is the shape of $c$: as the reviewer notes, a learned critic is not globally strongly convex, but this simple setting allows us to isolate the ReMax mechanism.
>
> To clarify the scope, Sec. 5.2 now states that Proposition 1 does not require convexity, whereas Propositions 2 and 3 use strong convexity.
> We also explicitly distinguish the fixed-state, fixed-critic analysis from ReMAC's learned critic and coupled actor--critic dynamics.
>
>
> ### W2. Entropy, state coverage, and sparse rewards
>
> Our scope is **stochastic exploration** induced by policy stochasticity, for which policy entropy is a direct measure. That said, to address the questions, we added evaluations of state-space coverage and sparse-reward performance.
>
> **For coverage**, following maximum state-entropy exploration methods (MEPOL [2], RE3 [3]), we report a $k$-nearest-neighbor estimate of visited-state entropy in App. C.6. ReMAC does not consistently outperform SAC. However, larger $M$ often yields higher estimates than $M=1$, which has no exploration bonus, providing additional evidence that ReMax promotes exploration. Specifically, $M=2$ and $M=4$ outperform $M=1$ on HalfCheetah, while $M=4$ outperforms $M=1$ on Hopper. On Ant, ReMAC is comparable to SAC within the across-seed variation.
>
> **For sparse rewards**, following JaxGCRL [4], we replace Reacher’s shaped reward with $r_t=\mathbf{1} [d_t<0.01]$ in App. C.7. Every $M>1$ improves over $M=1$ ($24.0\pm8.9$), achieving $30.7\pm5.0$, $31.2\pm4.8$, and $28.0\pm5.0$ for $M=2,4,8$, respectively, while remaining comparable to SAC ($29.5\pm5.4$); a uniformly random policy scores $0.15$ (out of $50$).
>
> Overall, as we have claimed in the paper, harder exploration settings may still require temporally coherent or uncertainty-directed exploration beyond the mechanism studied here, and we identify them as important future work, particularly in combination with a posterior over the $Q$-function.
>
> ### W3. Computational cost
> ReMAC evaluates $B$ actor samples per state, and Tab. 3 shows that $B=16$ roughly doubles the wall-clock time relative to SAC. The batch-size ablation shows similar learning curves for $B=8$ and $B=16$, so the cheaper setting is usually sufficient. Since performance is not monotone in $M$, we recommend $M\in\{2,4\}$ with a small $B$ as practical defaults.
>
> **Reference**
> [1] S. Bubeck.
> Convex Optimization: Algorithms and Complexity.
> Foundations and Trends in Machine Learning, 8(3--4):231--357, 2015.
> [2] M. Mutti et al.
> Task-Agnostic Exploration via Policy Gradient of a Non-Parametric State Entropy Estimate.
> AAAI Conference on Artificial Intelligence, 2021.
> [3] Y. Seo et al.
> State Entropy Maximization with Random Encoders for Efficient Exploration.
> International Conference on Machine Learning (ICML), 2021.
> [4] M. Bortkiewicz et al.
> Accelerating Goal-Conditioned Reinforcement Learning Algorithms and Research.
> International Conference on Learning Representations (ICLR), 2025.

---

> ### Author Response · Authors · 2026-07-15
>
> ### Q1. Does performance increase with $M$? How should $M$ be chosen?
> Performance is not monotone in $M$, and the best value depends on the environment, while $M>1$ generally matches or improves over $M=1$, while maintaining higher policy entropy in the six main tasks. We therefore treat $M$ as a stochasticity-control hyperparameter and recommend $M\in\{2,4\}$ as practical defaults given the constraint $B\ge M$.
>
> ### Q2. Bound the Proposition 3 expectations in $\mu$, $\sigma$, and $M$
> Following the suggestion, we added Corollary 1, which bounds the two expectation terms in Proposition 3 explicitly in terms of $\mu$, $\sigma$, and $M$.
> The resulting bound on the mean gradient decreases with $M$, making the gradient-damping effect more explicit.
> For the scale gradient, the nearest-sample term also decreases with $M$, although the full bound is not necessarily monotone because of an additional slowly growing factor.
> **These results clarify the dependence of the bounds on the retry budget.**
> We thank the reviewer for this helpful suggestion.
>
>
> ### Q3. Is $\epsilon_{1:B}$ scalar or vector?
> It is vector-valued. For $d$-dimensional actions, $\xi_i\sim\mathcal{N}(0,I_d)$ and
> $\tilde a_i=\mu_\theta(s)+\sigma_\theta(s)\odot\xi_i.$
> Secs. 4.2 and 6 now state this explicitly. We also renamed the Gaussian noise from $\epsilon$ to $\xi$ to avoid confusion with Adam's stabilization constant.
>
> ### Typo: $Z_m$ is undefined
> Fixed. We now define
> $Z_1,\ldots,Z_M \overset{\mathrm{iid}}{\sim} \mathcal{N}(0,1), \qquad \beta_M=\mathbb{E}\left[\max_{m\le M}Z_m\right]$
> at first use.
>
> We thank the reviewer again for the constructive feedback, which helped us substantially improve the paper.